# Learning Grid Cells as Vector Representation of Self-Position Coupled with Matrix Representation of Self-Motion

**Ruiqi Gao**[1]*, **Jianwen Xie**[2]*, **Song-Chun Zhu**[1] **& Ying Nian Wu**[1]
[1]University of California, Los Angeles, USA
[2]Hikvision Research Institute, Santa Clara, USA
{ruiqigao, jianwen}@ucla.edu, {sczhu, ywu}@stat.ucla.edu

## Abstract

This paper proposes a representational model for grid cells. In this model, the 2D self-position of the agent is represented by a high-dimensional vector, and the 2D self-motion or displacement of the agent is represented by a matrix that transforms the vector. Each component of the vector is a unit or a cell. The model consists of the following three sub-models. (1) Vector-matrix multiplication. The movement from the current position to the next position is modeled by matrix-vector multiplication, i.e., the vector of the next position is obtained by multiplying the matrix of the motion to the vector of the current position. (2) Magnified local isometry. The angle between two nearby vectors equals the Euclidean distance between the two corresponding positions multiplied by a magnifying factor. (3) Global adjacency kernel. The inner product between two vectors measures the adjacency between the two corresponding positions, which is defined by a kernel function of the Euclidean distance between the two positions. Our representational model has explicit algebra and geometry. It can learn hexagon patterns of grid cells, and it is capable of error correction, path integral and path planning.

## 1 Introduction

Imagine you are walking in your living room in the dark at night without any visual cues. Purely based on your self-motion, you know where you are while you are walking simply by integrating your self-motion. This is called path integral (Hafting et al. (2005); Fiete et al. (2008); McNaughton et al. (2006)). You can also plan your path to the light switch or to the door. This is called path planning (Fiete et al. (2008); Erdem & Hasselmo (2012); Bush et al. (2015)). You need to thank your grid cells for performing such navigation tasks.

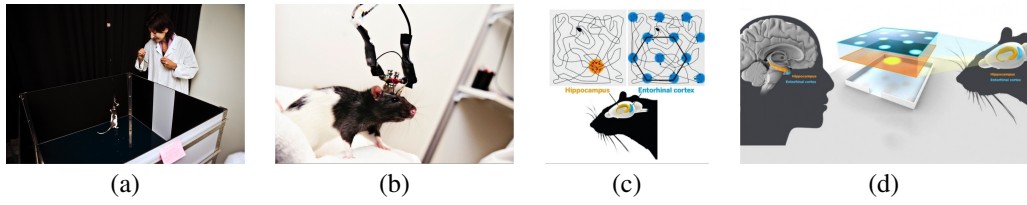

<p align="center">(a)      (b)      (c)      (d)</p>

Figure 1: Place cells and grid cells. (a) The rat is moving within a square region. (b) The activity of a neuron is recorded. (c) When the rat moves around (the curve is the trajectory), each place cell fires at a particular location, but each grid cell fires at multiple locations that form a hexagon grid. (d) The place cells and grid cells exist in the brains of both rat and human. (Source of pictures: internet)

Figure 1(a) shows Dr. May-Britt Moser, who together with Dr. Edvard Moser, won the 2014 Nobel Prize for Physiology or Medicine, for their discovery of the grid cells (Hafting et al. (2005); Fyhn et al. (2008); Yartsev et al. (2011); Killian et al. (2012); Jacobs et al. (2013); Doeller et al. (2010))

---

*Equal contributions.

in 2005. Their thesis advisor, Dr. John O'keefe, shared the prize for his discovery of the place cells (O'Keefe (1979)). Both the place cells and grid cells are used for navigation. The discoveries of these cells were made by recording the activities of the neurons of the rat when it moves within a square region. See Figure 1(b). Some neurons in the Hippocampus area are place cells. Each place cell fires when the rat moves to a particular location, and different place cells fire at different locations. The whole collection of place cells cover the whole square region. The discovery of grid cells was much more surprising and unexpected. The grid cells exist in the Entorhinal cortex. Each grid cell fires at multiple locations, and these locations form a regular hexagon grid. See Figure 1(c). The grid cells have been discovered across mammalian species, including human. See Figure 1(d).

In this paper, we propose a representational model to explain the hexagon patterns of the grid cells, and to explain how the grid cells perform path integral and path planning. We shall show that the grid cells are capable of error correction, which provides a justification for the grid cells.

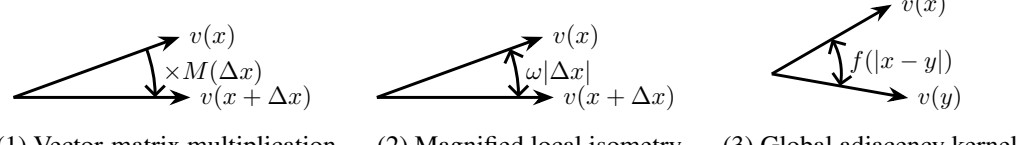

(1) Vector-matrix multiplication     (2) Magnified local isometry     (3) Global adjacency kernel

Figure 2: Grid cells form a high-dimensional vector representation of 2D self-position. Three sub-models: (1) Local motion is modeled by vector-matrix multiplication. (2) Angle between two nearby vectors magnifies the Euclidean distance. (3) Inner product between any two vectors measures the adjacency which is a kernel function of the Euclidean distance.

Figure 2 illustrates our model. The 2D self-position of the agent is represented by a high-dimensional vector, and the 2D self-motion or displacement of the agent is represented by a matrix that acts on the vector. Each component of the vector is a unit or a cell. In Figure 2, $x$ denotes the 2D self-position, $v(x)$ is the high-dimensional vector representation of the 2D $x$. $\Delta x$ is the self-motion or one-step displacement. $M(\Delta x)$ is the matrix representation of $\Delta x$. The model consists of the following three sub-models. (1) Vector-matrix multiplication. The movement from the current position $x$ to the next position $x + \Delta x$ is modeled by matrix-vector multiplication, i.e., the vector of the next position $v(x + \Delta x)$ is obtained by multiplying the matrix of the motion, $M(\Delta x)$, to the vector of the current position $v(x)$. (2) Magnified local isometry. The angle between two nearby vectors equals the Euclidean distance $|\Delta x|$ between the two corresponding positions multiplied by a magnifying factor. (3) Global adjacency kernel. The inner product between two vectors $\langle v(x), v(y) \rangle$ measures the adjacency between the two corresponding positions, which is defined by a kernel function $f$ of the Euclidean distance $|x - y|$ between the two positions $x$ and $y$. One additional feature is that the whole vector $v(x)$ is partitioned into multiple sub-vectors, and each sub-vector is driven by an associated sub-matrix. The whole system is like a multi-arm clock, with each arm rotating at a magnified speed and spanning a 2D sub-manifold on a high-dimensional sphere.

Our experiments show that sub-models (1) and (2) are sufficient for the emergence of the hexagon grid patterns of the grid cells. Sub-model (2) makes the vector representation robust to noises or errors due to the magnification of the distance. Sub-model (3) enables unique decoding of the position from the vector, because $f$ is unimodal and peaked at 0, and it may serve as the link between the grid cells and place cells. Together with sub-model (1), sub-model (3) enables path integral and path planning, because the adjacency $\langle v(x), v(y) \rangle$ informs the Euclidean distance $|x - y|$. All the three sub-models can be implemented by one-layer neural networks.

## 2   CONTRIBUTIONS AND RELATED WORK

The following are the contributions of our work. (1) We propose a representational model for grid cells, where the self-position is represented by a vector and the self-motion is represented by a matrix that acts on the vector. (2) We show that our model can learn hexagon grid patterns. (3) We show our model is capable of path integral, path planning, and error correction.

Many mathematical and computational models (Burak & Fiete (2009); Sreenivasan & Fiete (2011); Blair et al. (2007); de Almeida et al. (2009)) have been proposed to explain the formation and function of grid cells. Compared to previous computational models on grid cells, our model only makes very generic assumptions about the algebra and geometry of the representational scheme, without assuming Fourier plane waves or clock arithmetics.

Recently, deep learning models (Banino et al. (2018); Cueva & Wei (2018)) have been proposed to learn the grid-like units for navigation. Our work was inspired by them. Compared with these models, our model has explicit algebra and geometry. In terms of algebra, our model has explicit matrix representation of self-motion, and the change of self-position is modeled by vector-matrix multiplication. In terms of geometry, our model assumes that the vector rotates while the agent moves, and our model assumes magnified local isometry and global adjacency kernel based on the angles between the vectors.

Expressing the adjacency kernel as the inner product between the vectors is related to the kernel trick (Cortes & Vapnik (1995)), random Fourier basis (Ng et al. (2002)), and spectral clustering (Ng et al. (2002)).

## 3 REPRESENTATIONAL MODEL OF GRID CELLS

Consider an agent navigating within a domain $D = [0, 1] \times [0, 1]$ (actually the shape does not matter, and it can be $\mathbb{R}^2$). We can discretize $D$ into an $N \times N$ lattice. $N = 40$ in our experiments. Let $x = (x_1, x_2) \in D$ be the self-position of the agent. $x$ is 2D. Suppose the agent wants to represent its self-position by a $d$-dimensional hidden vector $v(x)$. We introduce the following three sub-models.

### 3.1 SUB-MODEL 1 ABOUT MOTION ALGEBRA: VECTOR-MATRIX MULTIPLICATION

Suppose at a position $x$, the self-motion or one-step displacement is $\Delta x$, so that the agent moves to $x + \Delta x$ after one step. We assume that

$$v(x + \Delta x) = M(\Delta x)v(x), \tag{1}$$

where $M(\Delta x)$ is a $d \times d$ matrix that depends on $\Delta x$. While $v(x)$ is the vector representation of the self-position $x$, $M(\Delta x)$ is the matrix representation of the self-motion $\Delta x$. We can illustrate the motion model by the following diagram:

$$
\begin{array}{ccc}
\text{Motion}: & x_t & \xrightarrow{\ +\Delta x\ } & x_{t+1} \\
& \downarrow & \downarrow & \downarrow \\
& v(x_t) & \xrightarrow{\ M(\Delta x)\times\ } & v(x_{t+1})
\end{array}
\tag{2}
$$

See Figure 2(1). Both $v(x)$ and $M(\Delta x)$ are to be learned.

We can discretize $\Delta x$, and learn a motion matrix $M$ for each $\Delta x$. We can also learn a parametric model for $M$. To this end, we can further parametrize $M = I + \tilde{M}(\Delta x)$ such that each element of $\tilde{M}(\Delta x)$ is a quadratic (or polynomial) function of $\Delta x = (\Delta x_1, \Delta x_2)$:

$$\tilde{M}_{ij}(\Delta x) = \beta_{ij}^{(1)}\Delta x_1 + \beta_{ij}^{(2)}\Delta x_2 + \beta_{ij}^{(11)}\Delta x_1^2 + \beta_{ij}^{(22)}\Delta x_2^2 + \beta_{ij}^{(12)}\Delta x_1\Delta x_2, \tag{3}$$

where the coefficients $\beta$ are to be learned. The above may be considered a second-order Taylor expansion which is expected to be accurate for small $\Delta x \in \Delta$, where $\Delta$ is the allowed collection of one-step displacements, e.g., $\pm 3$ grid points in each direction on the $40 \times 40$ grid.

The motion model can be considered a linear recurrent neural network (RNN). However, if we are to interpret $M$ as the weight matrix, then the weight matrix is dynamic because it depends on the motion $\Delta x$. One may implement it by discretizing $\Delta x$, so that we have a finite set of $\Delta x$, and thus a finite set of $M(\Delta x)$. Then at each time step, the RNN switches between the finite set of motion matrices. This is like the gearing operation of a multi-speed bicycle.

### 3.2 DISENTANGLED BLOCKS OR MODULES

For the sake of estimation accuracy and computational efficiency, we further assume that $M(\Delta x)$ is block diagonal, i.e., we can divide $v$ into $K$ blocks of sub-vectors, $v = (v^{(k)}, k = 1, ..., K)$, and

$v^{(k)}(x + \Delta x) = M^{(k)}(\Delta x)v^{(k)}(x)$. That is, the vector $v$ consists of sub-vectors, each of which rotates in its own subspace, so that the dynamics of the sub-vectors are disentangled.

The assumption of disentangled blocks in our model is related to the modular organization of grid cells in neuroscience (Stensola et al. (2012)), where a module refers to a region of grid cells where all cells share similar grid scales and orientations.

### 3.3 SUB-MODEL 2 ABOUT LOCAL GEOMETRY: MAGNIFIED LOCAL ISOMETRY

The above motion algebra alone is not sufficient for learning the vector matrix representations, because a trivial solution is that all the $v(x)$ are the same, and the matrix $M(\Delta x)$ is always identity. We need to properly displace the vectors $v(x)$. So we shall model $\langle v(x), v(y) \rangle$ both locally and globally.

Let $d$ be the dimensionality of $v^{(k)}$, i.e., the number of grid cells within the $k$-th block. For the local geometry, we assume that for each block,

$$\langle v^{(k)}(x), v^{(k)}(x + \Delta x) \rangle = d(1 - \alpha_k|\Delta x|^2), \tag{4}$$

for all $x$ and $\Delta x$ such that $\alpha_k|\Delta x|^2 \leq c$, i.e., $\Delta x \in \Delta(\alpha_k) = \{\Delta x : \alpha_k|\Delta x|^2 \leq c\}$. In our experiments, we take $c = 1.5$. $\alpha_k$ can be either designed or learned.

Based on sub-model (4), $\|v^{(k)}(x)\|^2 = d$ for every $x$. The inner product on the left hand side is $d\cos(\Delta\theta)$ where $\Delta\theta$ is the angle between $v^{(k)}(x)$ and $v^{(k)}(x + \Delta x)$. $1 - \alpha_k|\Delta x|^2$ on the right hand side may be considered a second order Taylor expansion of a function $f(r)$ such that $f(0) = 1$, $f'(0) = 0$, i.e., 0 is the maximum, and $f''(0) = -2\alpha_k$. It is also an approximation to $\cos(\sqrt{2\alpha_k}|\Delta x|)$. Let $\omega_k = \sqrt{2\alpha_k}$, we have

$$\text{Magnified local isometry} : \Delta\theta = \omega_k|\Delta x|, \tag{5}$$

i.e., the angle between $v(x)$ and $v(x + \Delta x)$ magnifies the distance $|\Delta x|$ by a factor of $\omega_k$ uniformly for all $x$. See Figure 2(2).

The factor $\omega_k$ defines the metric of block $k$.

### 3.4 ROTATION AND PROJECTION

Since $\|v^{(k)}(x)\|$ is a constant for all $x$, $M^{(k)}(\Delta x)$ is an orthogonal matrix, and the self-motion is represented by a rotation in the $d$-dimensional space. $\omega_k|\Delta x|$ is the angular speed of rotation of the sub-vector $k$. $(v^{(k)}(x), \forall x)$ forms a 2D sub-manifold on the sphere in the $d$-dimensional space. $v^{(k)}$ is like an arm of a clock except that the clock is not a 1D circle, but a 2D sub-manifold.

This 2D sub-manifold becomes a local codebook for the 2D positions within a local neighborhood. For a vector $v^{(k)}$, we can decode its position by projecting it onto the 2D sub-manifold, to get $\hat{x} = \arg\max_x \langle v^{(k)}, v^{(k)}(x) \rangle$ where the maximization is within the local neighborhood.

### 3.5 ERROR CORRECTION

The neurons are intrinsically noisy and error prone. For $\omega_k = \sqrt{2\alpha_k} \gg 1$, the magnification offers error correction because $v^{(k)}(x)$ and $v^{(k)}(x + \Delta x)$ are far apart, which is resistant to noises or corruptions. That is, projection to the 2D sub-manifold codebook removes the noises.

Specifically, suppose we have a vector $u = v^{(k)}(x) + \epsilon$, where $\epsilon \sim \mathrm{N}(0, s^2 I_d)$, and $I_d$ is the $d$-dimensional identity matrix. We can decode $x$ from $u$ based on the codebook by maximizing $\langle u, v^{(k)}(y) \rangle = \langle v^{(k)}(x), v^{(k)}(y) \rangle + \langle \epsilon, v^{(k)}(y) \rangle = d(1 - \alpha_k|y - x|^2) + \sqrt{d}sZ$ over $y$ that is within a local neighborhood of $x$, where $Z \sim \mathrm{N}(0, 1)$. The optimal $y$ will be close to $x$ because if $y$ deviates from $x$ by $\Delta x$, then the first term will drop by $d\alpha_k|\Delta x|^2$, which cannot be made up by the second term $\sqrt{d}sZ$ due to noises, unless the noise level $s$ is extremely large. From the above analysis, we can also see that the larger $d$ is, the more resistant the system is to noise, because the first term is of order $d$ and the second term is of order $\sqrt{d}$.

In addition to additive noises, the system is also resistant to multiplicative noises including dropout errors, i.e., multiplicative Bernoulli 0/1 errors. The dropout errors may occur due to noises, aging, or diseases. They may also be related to the asynchronous nature of the neuron activities in computing.

### 3.6 HEXAGON GRID PATTERNS

While the magnified local isometry enables error correction, it also causes global periodicity. Because the angle between nearby $v^{(k)}(x)$ and $v^{(k)}(x + \Delta x)$ is magnified, when we vary $x$, the vector $v^{(k)}(x)$ will rotate at a magnified speed $\omega_k |\Delta x|$, so that it quickly rotates back to itself, like an arm of a clock. Thus each unit of $v^{(k)}(x)$ is periodic with $\omega_k$ determining the periodicity.

Our experiments show that sub-models (1) and (2) are sufficient for the emergence of hexagon grid patterns. In fact, we have the following analytical solution (see Appendix for a proof):

**Theorem 1.** *Let $e(x) = (e^{i\langle a_j, x \rangle}, j = 1, 2, 3)^\top$, and $a_1$, $a_2$, $a_3$ are three 2D vectors so that the angle between $a_i$ and $a_j$ is $2\pi/3$ for $\forall i \neq j$ and $|a_j| = 2\sqrt{\alpha}$ for $\forall j$. Let $C$ be a random $3 \times 3$ complex matrix such that $C^* C = I$. Then $v(x) = Ce(x)$, $M(\Delta x) = C\mathrm{diag}(e(\Delta x))C^*$ satisfy equation 1 and equation 4 approximately for all $x$ and small $\Delta x$.*

$v(x)$ amounts to a 6-dimensional real vector. Since the angle between $a_i$ and $a_j$ is $2\pi/3$ for $\forall i \neq j$, patterns of $v(x)$ over $x$ have hexagon periodicity. Moreover, the scale of the patterns is controlled by the length of $a_j$, i.e., the scaling parameter $\alpha$.

We want to emphasize that sub-models (1) and (2) are about local $\Delta x$, where we do not make any assumptions about global patterns, such as Fourier basis. In contrast, the solution in the above theorem is global. That is, our model assumes much less than the solution in the theorem. Our experiments show that the hexagon patterns will emerge as long as the number of units is greater than or equal to 6.

### 3.7 SUB-MODEL 3 ABOUT GLOBAL GEOMETRY: ADJACENCY KERNEL

Because of the periodicity, each block $(v^{(k)}(x), \forall x)$ does not form a global codebook of 2D positions, i.e., there can be $x \neq y$, but $v^{(k)}(x) = v^{(k)}(y)$, i.e., $v^{(k)}$ does not encode $x$ uniquely. We can combine multiple blocks to resolve the global ambiguity. Specifically, let $v(x) = (v^{(k)}(x), k = 1, ..., K)$ be the whole vector, we assume the following global adjacency sub-model for the whole vector:

$$\langle v(x), v(y) \rangle = \sum_{k=1}^{K} \langle v^{(k)}(x), v^{(k)}(y) \rangle = (Kd)f(|x - y|), \tag{6}$$

where $(v(x), M(\Delta x), \alpha_k, \forall x, \Delta x, k)$ are to be learned.

Recall $d$ is the number of grid cells in each block, and $K$ is the number of blocks. $f(r)$ is the adjacency kernel that decreases monotonically as the Euclidean distance $r = |x - y|$ increases. One example of $f$ is the Gaussian kernel $f(r) = \exp\left(-r^2/2\sigma^2\right)$. Another example is the exponential kernel $f(r) = \exp\left(-r/\sigma\right)$. As a matter of normalization, we assume $f(0) = 1$, which is the maximum of $f(r)$.

Since $f(0) = 1$, $\|v(x)\|^2 = Kd$ for any $x$, and $\langle v(x), v(y) \rangle = (Kd)\cos\theta$, where $\theta$ is the angle between $v(x)$ and $v(y)$, and we have

$$\text{Global adjacency}: \cos\theta = f(|x - y|). \tag{7}$$

The angle between any two vectors is always less than $\pi/2$. See Figure 2(3).

By fitting the multiple sub-vectors together, we still retain the error correction capacity due to magnified local isometry, meanwhile we eliminate the ambiguity by letting $\langle v^{(k)}(x), v^{(k)}(y) \rangle$ for different $k$ cancel each other out by destructive interference as $y$ moves away from $x$, so that we obtain unique decoding of positions. Let $C = \{v(x), x \in D\}$ be the codebook sub-manifold, error correction of a vector $u$ is obtained by projection onto $C$: $\arg\max_{v \in C} \langle u, v \rangle$.

The whole vector $v$ is like a $K$-arm clock, with each $v^{(k)}$ being an arm rotating at a speed $\omega_k |\Delta x| = \sqrt{2\alpha_k} |\Delta x|$.

### 3.8 LOCALIZATION AND HEAT MAP

$(v(x), \forall x)$ forms a global codebook for $x$. It is a 2D sub-manifold on the sphere in the $(Kd)$-dimensional space. For a vector $v$, we can decode its position by its projection on the codebook

manifold. Since $f(r)$ is monotonically decreasing, $h(x) = \langle v, v(x) \rangle$ gives us the heat map to decode the position of $v$ uniquely. Let the decoded position be $\hat{x}$, then $\hat{x} = \arg \max_x \langle v, v(x) \rangle$. We can obtain the one-hot representation $\delta_{\hat{x}}$ of $\hat{x}$ by non-maximum suppression on the heat map $h(x)$.

Let $V = (v(x), \forall x)$ be the $(Kd) \times N^2$ matrix (recall the domain $D = [0, 1]^2$ is discretized into an $N \times N$ lattice), where each column is a $v(x)$. We can write the heat map $h(x) = \langle v, v(x) \rangle$ as a $N^2$-dimensional vector $h = V^\top v$, which serves to decode the position $x$ encoded by $v$. Conversely, for a one-hot representation of a position $x$, i.e., $\delta_x$, which is a one-hot $N^2$-dimensional vector, we can encode it by $v = V \delta_x$. Both the encoder and decoder can be implemented by a linear neural network with connection weights $V$ and $V^\top$ respectively, as illustrated by the following diagram:

$$
\text{Localization}: \quad v \quad \xrightarrow{V^\top \times} \quad h \quad \text{(heat map and decoding to } \delta_x\text{)}
$$
$$
\delta_x \quad \xrightarrow{V \times} \quad v(x) \quad \text{(encoding)}
$$
(8)

Note that in decoding $v \to h(x) \to \delta_x$ and encoding $\delta_x \to v$, we do not represent or operate on the 2D coordinate $x$ explicitly, i.e., $x$ itself is never explicitly represented, although we may use the notation $x$ in the description of the experiments.

### 3.9  PATH INTEGRAL

Path integral (also referred to as dead-reckoning) is the task of inferring the self-position based on self-motion (e.g., imagine walking in a dark room). Specifically, the input to path integral is a previously determined initial position $x_0$ and motion sequences $\{\Delta x_1, ..., \Delta x_T\}$, and the output is the prediction of one's current position $x_T$. We first encode the initial position $x_0$ as $v(x_0)$. Then, by the motion model, the hidden vector $v(x_T)$ at time $T$ can be predicted as:

$$
v(x_T) = \prod_{t=T}^{1} M(\Delta x_t) v(x_0).
$$
(9)

We can then decode $x_T$ from $v(x_T)$.

### 3.10  PATH PLANNING

Our representation system can plan direct path from the starting position $x_0$ to the target position $y$ by steepest ascent on the inner product $\langle v(x), v(y) \rangle$, i.e., let $v_0 = v(x_0)$, the algorithm iterates

$$
\Delta x_t = \arg \max_{\Delta x \in \Delta} \langle v(y), M(\Delta x) v_{t-1} \rangle,
$$
(10)

$$
v_t = M(\Delta x_t) v_{t-1},
$$
(11)

where $\Delta$ is the set of allowable displacements $\Delta x$.

When a rat does path planning, even if it is not moving, its grid cells are expected to be active. This may be explained by the above algorithm. In path planning, the rat can also fantasize bigger step sizes that are beyond its physical capacity, by letting $\Delta$ include physically impossible large steps.

In general, our representation scheme $(v(x), M(\Delta x), f(|x - y|))$ mirrors $(x, \Delta x, |x - y|)$. Thus the learned representation is capable of implementing existing path planning algorithms in robotics (Siegwart et al. (2011)) even though our system does not have explicit coordinates in 2D (i.e., the 2D coordinates $x$ are never explicitly represented by two neurons).

### 3.11  PLACE CELLS AND SCALE

For each $x$, we may interpret $\langle v(x), v(y) \rangle / (Kd) = f(|x - y|)$ as a place cell whose response is $f(|x - y|)$ if the agent is at $y$, or at least, we may consider $f(|x - y|)$ as an internal input to the place cell based on self-motion, in addition to external visual cues.

For the Gaussian kernel $f(r) = \exp(-r^2/2\sigma^2)$, the choice of $\sigma$ determines the scale. In path integral, for accurate decoding of the position $x$ from the vector $v$ in the presence of noises, we want $\sigma$ to be small so that $f(r)$ drops to zero quickly. However, in path planning, for a vector $v$, we also want to know the Euclidean distance between its position $x$ to a target position $y$, which

may be far away from $x$. The distance is $|x - y| = f^{-1}(\langle v, v(y) \rangle / (Kd))$. For accurate estimation of long distance in the presence of noises, we need $\sigma$ to be large, so that the slope of $f^{-1}$ is not too big. Perhaps we need multiple $f(r)$ with different $\sigma$, and for each $f(r)$ we have $(Kd)f(r) = \sum_{k=1}^{K} \gamma_k \langle v^{(k)}(x), v^{(k)}(y) \rangle$, where different $f(r)$ have different coefficients $(\gamma_k)$ while sharing the same $(v^{(k)}(x))$. We shall study this issue in future work.

### 3.12 GROUP REPRESENTATION IN MOTOR CORTEX

Where does the self-motion $\Delta x$ come from? It comes from the movements of head and legs, i.e., each step of navigation involves a whole process of path integral and path planning of the movements of head and legs (as well as arms). In general, we can use the same system we have developed for the movements of head, legs, arms, fingers, etc. Their movements form groups of actions. In navigation, the movements belong to 2D Euclidean group. In body movements, the movements belong to various Lie groups. Our method can be used to learn the representational systems of these groups, and such representations may exist in motor cortex. We leave this problem to future investigation.

## 4 LEARNING REPRESENTATION

The square domain $D$ is discretized into a $40 \times 40$ lattice, and the agent is only allowed to move on the lattice. We can learn $(v(x), \forall x)$ and $(M(\Delta x), \forall \Delta x)$ (or the $\beta$ coefficients that parametrize $M$) by minimizing the following loss functions.

For sub-model (1) on vector-matrix multiplication, the loss function is

$$L_1 = \mathbb{E}_{x, \Delta x} \left[ \|v(x + \Delta x) - M(\Delta x)v(x)\|^2 \right], \tag{12}$$

where $x$ is sampled from the uniform distribution on $D = [0, 1]^2$, and $\Delta x$ is sampled from the uniform distribution within a certain range $\Delta_1$ (3 grid points in each direction in our experiments). The above motion loss is a single-step loss. It can be generalized to multi-step loss

$$L_{1,T} = \mathbb{E}_{x, \Delta x_1, \dots \Delta x_T} \left[ \|v(x + \Delta x_1 + \dots + \Delta x_T) - M(\Delta x_T) \dots M(\Delta x_1)v(x)\|^2 \right], \tag{13}$$

where $(\Delta x_t, t = 1, \dots, T)$ is a sequence of $T$ steps of displacements, i.e., a simulated trajectory.

For sub-model (2) on magnified local isometry, the loss function is

$$L_{2,k} = \mathbb{E}_{x, \Delta x} \left[ (\langle v^{(k)}(x), v^{(k)}(x + \Delta x) \rangle - (d(1 - \alpha_k |\Delta x|^2))^2 \right], \tag{14}$$

where for fixed $\alpha_k$, $\Delta x$ is sampled uniformly within the range $\Delta_2(\alpha_k) = \{\Delta x : \alpha_k |\Delta x|^2 \leq c\}$ ($c = 1.5$ in our experiments). We can define $L_2 = \sum_{k=1}^{K} L_{2,k}$.

For sub-model (3) on global adjacency kernel, the loss function is

$$L_3 = \mathbb{E}_{x, y} \left[ ((Kd)f(|x - y|) - \langle v(x), v(y) \rangle)^2 \right], \tag{15}$$

where both $x$ and $y$ are sampled uniformly from $D$.

Let $v(x) = (v_i(x), i = 1, \dots, Kd)$, we also impose a regularization loss $L_0 = \sum_{i=1}^{Kd} (\mathbb{E}_x[v_i(x)^2] - 1)^2$, to enforce uniform energy among the grid cell. It also helps to break the symmetry caused by the fact that the loss function is invariant under the transformation $v(x) \to Qv(x), \forall x$, where $Q$ is an arbitrary orthogonal matrix. This loss is not crucial though, so we will make it implicit for the rest of the paper.

Fixing the magnitude of $\mathbb{E}_x[v_i(x)]$ within a certain range is biologically plausible, because $v_i(x)$ is a single cell. For mathematical and computational convenience, we can also normalize $v(x)$ so that $\|v(x)\|^2 = 1$, $\langle v(x), v(y) \rangle = f(|x - y|)$, $\langle v^{(k)}(x), v^{(k)}(x + \Delta x) \rangle = (1 - \alpha_k |\Delta x|^2)/K$, and $\mathbb{E}_x[v_i(x)^2] = 1/(Kd)$. When learning a single block, we can normalize $v^{(k)}(x)$ so that $\|v^{(k)}(x)\|^2 = 1$, $\langle v^{(k)}(x), v^{(k)}(x + \Delta x) \rangle = 1 - \alpha_k |\Delta x|^2$ and $\mathbb{E}_x[v_i(x)^2] = 1/d$.

The total loss function is a linear combination of the above losses, where the weights for combining the losses are chosen so that the weighted losses are of the same order of magnitude.

The loss function is minimized by Adam optimizer (Kingma & Ba (2014)) (lr = 0.03) for $6,000$ iterations. A batch of $30,000$ examples, i.e., $(x, \Delta x_t, t = 1, ..., T))$ for $L_1$, $(x, \Delta x)$ for $L_2$, and $(x, y)$ for $L_3$ are sampled at each learning iteration as the input to the loss function. In the later stage of learning ($\geq 4,000$ iterations), $\|v(x)\| = 1$ is enforced by projected gradient descent, i.e., normalizing each $v(x)$ after the gradient descent step.

# 5 EXPERIMENTS

## 5.1 LEARNING SINGLE BLOCKS: HEXAGON PATTERNS AND METRICS

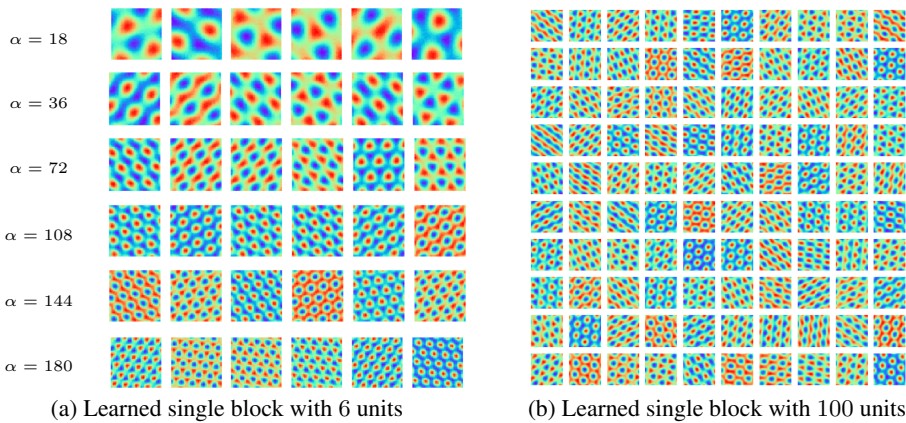

(a) Learned single block with 6 units    (b) Learned single block with 100 units

Figure 3: Learned units of a single block with fixed $\alpha$. (a) Learned single block with 6 units. Every row shows the learned units with a given $\alpha$. (b) Learned single block with 100 units and $\alpha = 72$.

We first learn a single block with fixed $\alpha_k$ by minimizing $L_1 + \lambda_2 L_{2,k}$ (we shall drop the subscript $k$ in this subsection for simplicity). Figure 3 shows the learned units over the $40 \times 40$ lattice of $x$. Figure 3(a) shows the learned results with 6 units and different values of $\alpha$. The scale or metric of the lattice is controlled by $\alpha$. The units within a block have patterns with similar scale and arrangement, yet different phases. Figure 3(b) shows the learned results with 100 units and $\alpha = 72$, indicating that the grid-like patterns are stable and easy to learn even when the number of units is large.

## 5.2 LEARNING MULTIPLE HEXAGON BLOCKS AND METRICS

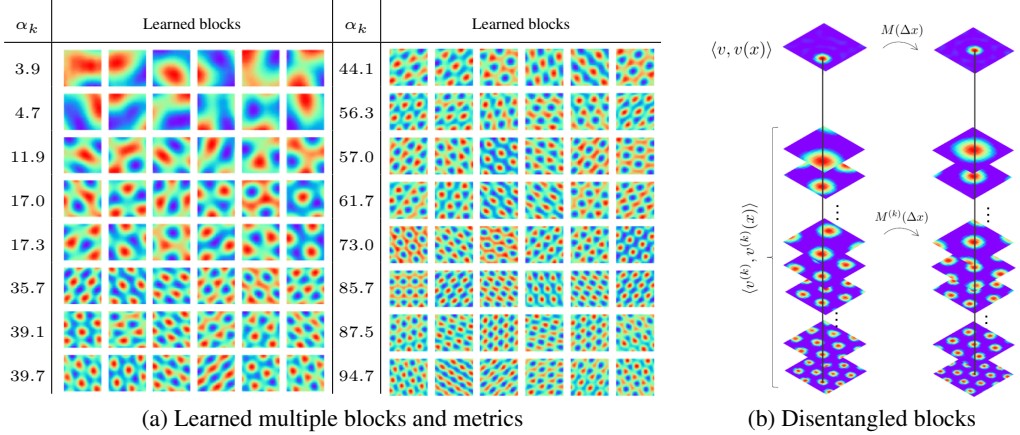

(a) Learned multiple blocks and metrics    (b) Disentangled blocks

Figure 4: (a) Response maps of learned units of the vector representation and learned scaling parameters $\alpha_k$. Block size equals 6 and each row shows the units belonging to the same block. (b) Illustration of block-wise activities of the units (where the activities are rectified to be positive).

We learn multiple blocks by minimizing $L_1 + \lambda_2 L_2 + \lambda_3 L_3$. Instead of manually assigning $\alpha_k$, we learn $\alpha_k$ by gradient descent, simultaneously with $v$ and $M$.

In Figure 4(a), we show the learned units $v(x)$ over the $40 \times 40$ lattice of $x$ and the learned metrics $\alpha_k$. A Gaussian kernel with $\sigma = 0.08$ is used for the global adjacency measure $f(|x - y|)$. Block size is set to 6 and each row shows the learned units belonging to the same block. The scales of the firing fields are controlled by the learned $\alpha_k$.

Figure 4(b) illustrates the combination of multiple blocks. For the localization model, given a vector $v$, the heat map of a single block $\langle v^{(k)}, v^{(k)}(x) \rangle$ has periodic firing fields and cannot determine a location uniquely. However, ambiguity disappears by combing the heat maps of multiple blocks, which have firing fields of multiple scales and phases that add up to a Gaussian kernel $\langle v, v(x) \rangle$. The Gaussian kernel informs the place cell, by which the location is determined uniquely. For a motion $\Delta x$, every block rotates in its own subspace with motion matrix $M^{(k)}(\Delta x)$, resulting in phase shifting in the heat map of each single block.

In the subsequent experiments, we shall learn the grid cells by minimizing $L_1 + \lambda_3 L_3$ for simplicity.

## 5.3 PATH INTEGRAL

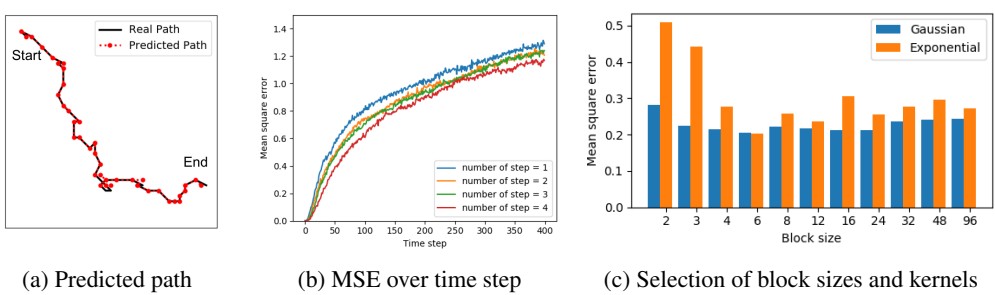

(a) Predicted path       (b) MSE over time step       (c) Selection of block sizes and kernels

Figure 5: (a) Path integral prediction. The black line depicts the real path while red dotted line is the predicted path by the learned model. (b) Mean square error over time step. The error is average over $1,000$ episodes. The curves correspond to different numbers of steps used in the multi-step motion loss. (c) Mean square error performed by models with different block sizes and different kernel types. Error is measured by number of grids.

Figure 5(a) shows an example of path integral result (time duration $\tilde{T} = 40$), where we use single step motion loss $L_{1, T=1}$ in learning. Gaussian kernel with $\sigma = 0.08$ is used as the adjacency measure in $L_3$. We find single-step loss is sufficient for performing path integral. The mean square error remains small ($\sim 1.2$ grid) even after $400$ steps of motions (figure 5(b)). The error is averaged over $1,000$ episodes. The motion loss can be generalized to multi-step $L_{1,T}$, as shown by equation 13. In Figure 5(b), we show that multi-step loss can improve the performance slightly.

In Figure 5(c) we compare the learned models with fixed number of units (96) but different block sizes. We also compare the performance of models using Gaussian kernel ($\sigma = 0.08$) and exponential kernel ($\sigma = 0.3$) as the adjacency measure in the localization model. The result shows that models with Gaussian kernel and block size $\geq 3$, and with exponential kernel and block size $\geq 4$ have performances comparable to the model learned without block-diagonal assumption (block size $= 96$).

## 5.4 PATH PLANNING

For path planning, we assume continuous $x$ and $\Delta x$. First, we design the set of allowable motions $\Delta$. For a small length $r$, we evenly divide $[0, 2\pi]$ into $n$ directions $\{\theta_i, i = 1, ..., n\}$, resulting in $n$ candidate motions $\Delta = \{\Delta x_i = (r \cos(\theta_i), r \sin(\theta_i)), i = 1, ..., n\}$. These $n$ small motions serve as motion basis. Larger motion can be further added to $\Delta$ by estimating the motion matrix $M(k \Delta x_i) = M^k(\Delta x_i)$. The starting position and destination can also be any continuous values, where the encoding to the latent vector is approximated by bilinear interpolation of nearest neighbors on the lattice.

In the experiments, we choose $r = 0.05$ and $n = 100$, and add another set of motions with length 0.025 to enable accurate planning. The system is learned with exponential kernel ($\sigma = 0.3$) as global adjacency to encourage connection of long distance. Figure 6(a) shows planning examples with six settings of motion ranges $\Delta$. Including larger motions accelerates the planning process so that it finishes with less steps. We define one episode to be a success if the distance between the end position and the destination is less than 0.025. We achieve a success rate of $> 99\%$ over $1,000$ episodes for all the six settings.

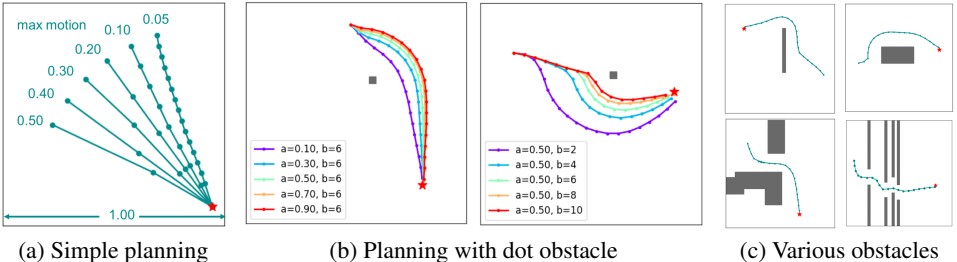

| (a) Simple planning | (b) Planning with dot obstacle | (c) Various obstacles |

Figure 6: (a) Planning examples with different motion ranges. Red star represents the destination $y$ and green dots represent the planned position $\{x_0 + \sum_{i=1}^{t} \Delta x_i\}$. (b) Planning examples with a dot obstacle. Left figure shows the effect of changing scaling parameter $a$, while right figure shows the effect of changing annealing parameter $b$. (c) Planning examples with obstacles mimicking walls, large objects and simple mazes.

The learned system can also perform planning with obstacles, where the global adjacency between the agent's current position and an obstacle serves as a repulsive forces. Specifically, suppose $z$ is an obstacle to avoid, the agent can choose the motion $\Delta x_t$ at time $t$ by

$$\Delta x_t = \arg \max_{\Delta x \in \Delta} \left[ \langle v(y), M(\Delta x) v_t \rangle - a \langle v(z), M(\Delta x) v_t \rangle^b \right], \tag{16}$$

where $a$ and $b$ are the scaling and annealing parameters. Figure 6(b) shows the planning result with a dot obstacle laid on the direct path between the starting position and destination, with tuning of $a$ and $b$. We choose $a = 0.5$ and $b = 6$ in subsequent experiments.

Now suppose we have more complicated obstacles $\{z_i\}_{i=1}^{m}$. They can be included by summing over the kernels of every obstacle $\{a \langle v(z_i), M(\Delta x) v_t \rangle^b\}_{i=1}^{m}$ and choosing $\Delta x_t$ at time $t$ by $\Delta x_t = \arg \max_{\Delta x \in \Delta} \left[ \langle v(y), M(\Delta x) v_t \rangle - \sum_{i=1}^{m} a \langle v(z_i), M(\Delta x) v_t \rangle^b \right]$. Figure 6(c) shows some examples, where the obstacles mimicking walls, large objects and simple mazes.

The above method is related to the potential field method in robotics (Siegwart et al. (2011)).

## 6 DISCUSSION: ROTATIONIST-CONNECTIONIST MODEL?

In terms of general modeling methodology, a typical recurrent network is of the form $v_t = \tanh(W(v_{t-1}, \delta_t))$, where $v_t$ is the latent vector, $\delta_t$ is the input change or action. $v_{t-1}$ and $\delta_t$ are concatenated and linearly mixed by $W$, followed by coordinate-wise tanh non-linearity. We replace it by a model of the form $v_t = M(\delta_t) v_{t-1}$, where $M(\delta_t)$ is a matrix that is non-linear in $\delta_t$. $M(\delta_t)$ is a matrix representation of $\delta_t$. $v_t$ can be interpreted as neuron activities. We can discretize the value of $\delta_t$ into a finite set $\{\delta\}$, and each $M(\delta)$ can be stored as synaptic connection weights that drive the neuron activities. In prediction, the input $\delta_t$ activates $M(\delta_t)$. In planning or control, all the $\{M(\delta)\}$ are activated, and among them the optimal $\delta$ is chosen.

The matrix representation of the above model is inspired by the group representation theory, where the group elements are represented by matrices acting on the vectors (Fulton & Harris (2013)). It underlies much of modern mathematics and holds the key to the quantum theory (Zee (2016)). Perhaps it also underlies the visual and motor cortex, where neurons form rotating sub-vectors driven by matrices representing groups of transformations. One may call it a rotationist-connectionist model.

## PROJECT PAGE

`http://www.stat.ucla.edu/~ruiqigao/gridcell/main.html`

## ACKNOWLEDGEMENT

We thank the three reviewers for their insightful comments and suggestions. Part of the work was done while Ruiqi Gao was an intern at Hikvision Research Institute during the summer of 2018. She thanks Director Jane Chen for her help and guidance. We also thank Jiayu Wu for her help with experiments and Zilong Zheng for his help with visualization. The work is supported by DARPA XAI project N66001-17-2-4029; ARO project W911NF1810296; ONR MURI project N00014-16-1-2007; and a Hikvision gift to UCLA. We gratefully acknowledge the support of NVIDIA Corporation with the donation of the Titan Xp GPU used for this research.

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

## A    PROOF FOR SECTION 3.6

*Proof.* $(a_j, j = 1, 2, 3)$ forms a tight frame in the 2D space, in that for any vector $x$ in 2D, $\sum_{j=1}^{3} \langle x, a_j \rangle^2 \propto |x|^2$.

Since $C^*C = I$, we have

$$
\begin{align}
\langle v(x), v(y) \rangle &= v(x)^* v(y) \tag{17} \\
&= e(x)^* C^* C e(y) \tag{18} \\
&= \sum_{j=1}^{3} e^{i\langle a_j, y-x \rangle}. \tag{19}
\end{align}
$$

Then we have

$$
\begin{align}
\mathrm{RE}(\langle v(x), v(x + \Delta x) \rangle) &= \sum_{j=1}^{3} \cos(\langle a_j, \Delta x \rangle) \tag{20} \\
&\approx \sum_{j=1}^{3} (1 - \langle a_j, \Delta x \rangle^2 / 2) \tag{21} \\
&= 3 - \sum_{j=1}^{3} \langle a_j, \Delta x \rangle^2 / 2 \tag{22} \\
&= 3(1 - \alpha |\Delta x|^2), \tag{23}
\end{align}
$$

where $|a_j| = 2\sqrt{\alpha}$.

The self motion from $v(x)$ to $v(x + \Delta x)$ is

$$
\begin{align}
v(x + \Delta x) &= C e(x + \Delta x) \tag{24} \\
&= C D(\Delta x) e(x) \tag{25} \\
&= C D(\Delta x) C^* v(x) \tag{26} \\
&= M(\Delta x) v(x), \tag{27}
\end{align}
$$

where $D(\Delta x) = \mathrm{diag}(e^{i\langle a_j, \Delta x \rangle}, j = 1, 2, 3)$. $\qquad\square$

If $K > 1$ and block size $= 6$, we can fit the multiple blocks together by a Fourier expansion of the kernel function

$$
\begin{align}
f(|x - y|) &= \langle v(x), v(y) \rangle \tag{28} \\
&\approx \sum_{k=1}^{K} \sum_{j=1}^{3} e^{i\langle a_{kj}, y-x \rangle} \tag{29} \\
&= \sum_{k=1}^{K} \langle v^{(k)}(x), v^{(k)}(y) \rangle. \tag{30}
\end{align}
$$

## B    HEXAGON GRID PATTERNS AND METRICS

### B.1    SIMULATED INPUT DATA

We obtain input data for learning the model by simulating agent trajectories with the number of steps equal to $\tilde{T}$ in the square domain $D$. $D$ is discretized into a $40 \times 40$ lattice and the agent is only allowed to move on the lattice. The agent starts at a random location $x_0$. At each time step $t$, a small motion $\Delta x_t$ ($\le 3$ grids in each direction) is randomly sampled with the restriction of not leading the agent outside the boundary, resulting in a simulated trajectory $\{x_0 + \sum_{i=1}^{t} \Delta x_i\}_{t=1}^{\tilde{T}}$. $\tilde{T}$ is set to $1,000$ to obtain trajectories that are uniformly distributed over the whole area. Although the trajectories used for training are restricted to the lattice and with small motions, in Section 5.4

we show that the learned model can be easily generalized to handle continuous positions and large motions. In training, pairs of locations $(x, y)$ are randomly sampled from each trajectory as the input to the adjacency loss $L_3$, while consecutive position sequences $(x, \Delta x_t, t = 1, ..., T)$ are randomly sampled as the input to the motion loss $L_1$, with length specified by $T$ (which is usually much smaller than the whole length of the trajectory $\tilde{T}$).

## B.2    LEARNED SINGLE BLOCK UNITS WITH DIFFERENT BLOCK SIZES

In (Blair et al. (2007)), the grid cell response is modeled by three cosine gratings with different orientations and phases. In our model, we learn such patterns of different scales without inserting artificial assumptions.

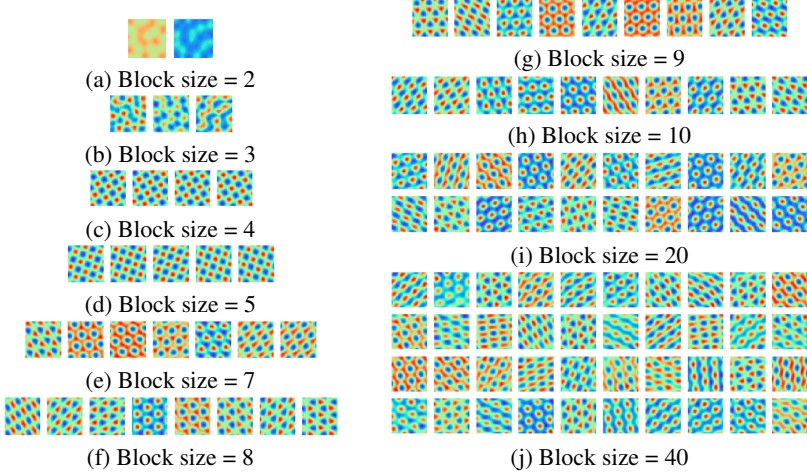

(a) Block size = 2

(b) Block size = 3

(c) Block size = 4

(d) Block size = 5

(e) Block size = 7

(f) Block size = 8

(g) Block size = 9

(h) Block size = 10

(i) Block size = 20

(j) Block size = 40

Figure 7: Response maps of learned single block units with different block sizes

Figure 7 displays the response maps of the learned single block units with different block sizes. For block sizes 4 and 5, the learned maps show square lattice patterns. For block sizes greater than or equal to 6, the learned maps show hexagon lattice patterns.

## B.3    LEARNED MULTIPLE BLOCK UNITS AND METRICS

### B.3.1    WITH DIFFERENT BLOCK SIZES

Figure 8 displays the response maps of the multiple block units with different block sizes. The metrics of the multiple blocks are learned automatically.

### B.3.2    WITH DIFFERENT SHAPES OF AREA

Figure 9 displays the response maps of the multiple block units with different shapes of the area, such as circle and triangle.

## B.4    QUANTITATIVE ANALYSIS OF SPATIAL ACTIVITY

We assess the spatial activity of the learned units quantitatively using measures adopted from the neuroscience literature. Specifically, we quantify the hexagonal regularity of the grid-like patterns using the gridness score (Langston et al. (2010); Sargolini et al. (2006)). The measure is derived from the spatial autocorrelogram of each unit's response map. A unit is classified as a grid cell if its gridness score is larger than 0. For those units that are classified as grid cells, grid scale and orientation can be further derived from the autocorrelogram following Sargolini et al. (2006). Figure 10(a) summarizes the results. 76 out of 96 learned units are classified as grid cells. Most units with large learned $\alpha_k$ are classified as grid cells, while those units with small $\alpha_k$ are not due to the lack of a full period of hexagonal patterns. The grid scales and orientations vary among units, while

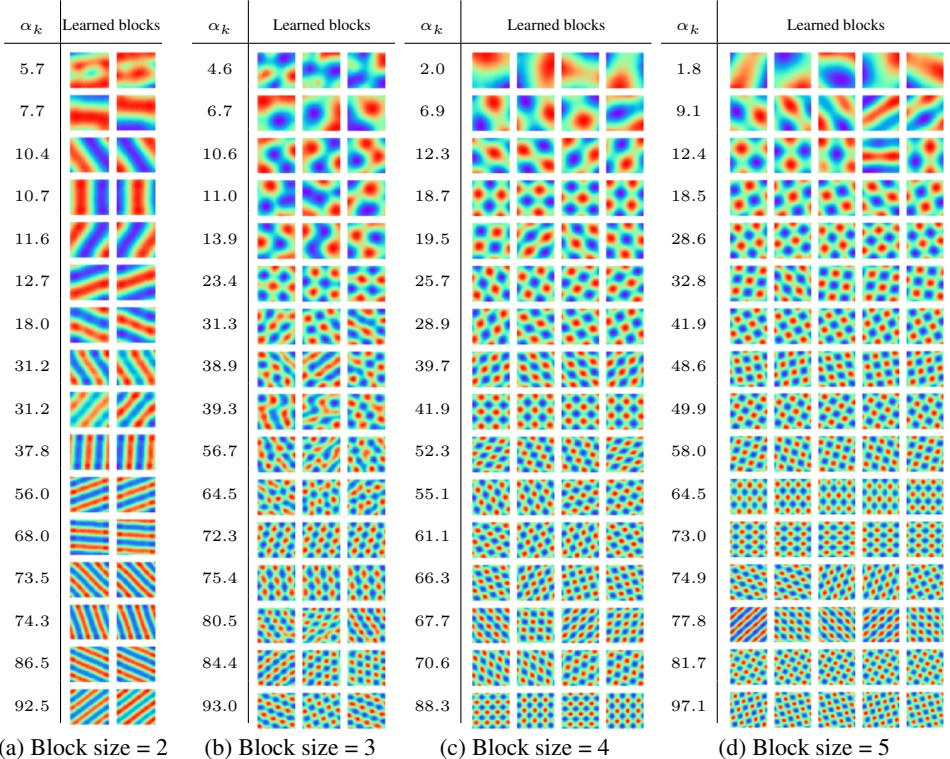

| $\alpha_k$ | Learned blocks | $\alpha_k$ | Learned blocks | $\alpha_k$ | Learned blocks | $\alpha_k$ | Learned blocks |
|---|---|---|---|---|---|---|---|
| 5.7 | | 4.6 | | 2.0 | | 1.8 | |
| 7.7 | | 6.7 | | 6.9 | | 9.1 | |
| 10.4 | | 10.6 | | 12.3 | | 12.4 | |
| 10.7 | | 11.0 | | 18.7 | | 18.5 | |
| 11.6 | | 13.9 | | 19.5 | | 28.6 | |
| 12.7 | | 23.4 | | 25.7 | | 32.8 | |
| 18.0 | | 31.3 | | 28.9 | | 41.9 | |
| 31.2 | | 38.9 | | 39.7 | | 48.6 | |
| 31.2 | | 39.3 | | 41.9 | | 49.9 | |
| 37.8 | | 56.7 | | 52.3 | | 58.0 | |
| 56.0 | | 64.5 | | 55.1 | | 64.5 | |
| 68.0 | | 72.3 | | 61.1 | | 73.0 | |
| 73.5 | | 75.4 | | 66.3 | | 74.9 | |
| 74.3 | | 80.5 | | 67.7 | | 77.8 | |
| 86.5 | | 84.4 | | 70.6 | | 81.7 | |
| 92.5 | | 93.0 | | 88.3 | | 97.1 | |
| (a) Block size = 2 | | (b) Block size = 3 | | (c) Block size = 4 | | (d) Block size = 5 | |

remaining similar within each block. We show the histograms of the grid orientations and scales in Figure 10(b) and 10(c) respectively. Moreover, we average the grid scales within each block and make the scatter plot of the averaged grid scales and the learned $1/\sqrt{\alpha_k}$ in figure 10(d). Interestingly, the grid scale is nicely proportional to the learned $1/\sqrt{\alpha_k}$.

## B.5 ABLATION STUDIES

We conduct ablation studies to assess various assumptions in our model.

### B.5.1 LOSS TERMS

We learn $v$ and $M$ with multiple blocks by minimizing $L_1 + \lambda_2 L_2 + \lambda_3 L_3$, which consists of (1) a motion loss $L_1$, (2) a local isometry loss $L_2$, and (3) a global adjacency loss $L_3$. Figure 11 shows the learned units when using only some of the components. Grid-like patterns do not emerge if using only the adjacency loss $L_3$, or only the isometry loss $L_2$. If using only the motion loss, the motion equation is approximately satisfied at the beginning of training, since both $v$ and $M$ are initialized from small values that are close to $0$. The system cannot be learned without an extra term to push $v(x)$ apart from each other. Figure 11(c) shows the learned units using the adjacency loss and the motion loss, leaving out the isometry loss. Grid-like patterns still emerge (also some strip-like patterns), although less obvious than the ones learned using the full loss.

### B.5.2 ASSUMPTIONS OF MOTION MATRIX

In another ablation study, we drop the quadratic parametrization by $\beta$ coefficients and the block diagonal assumption of the motion matrix. A separate motion matrix $M(\Delta x)$ is learned for each displacement $\Delta x$, and $v(x)$ is assumed to be a single block. We use either the local isometry loss $L_2$ or the global adjacency loss $L_3$ in addition to the motion loss $L_1$. The results are shown in Figure 12. With the isometry loss, the setting is similar to the one of learning single block, except that the parametrization of $M(\Delta x)$ is dropped. As shown in figure 12(a), the learned units resemble the ones learned with parametrized $M(\Delta x)$, but when the block size is large, the grid-like patterns are less obvious. With the adjacency loss, grid-like patterns do not emerge any more.

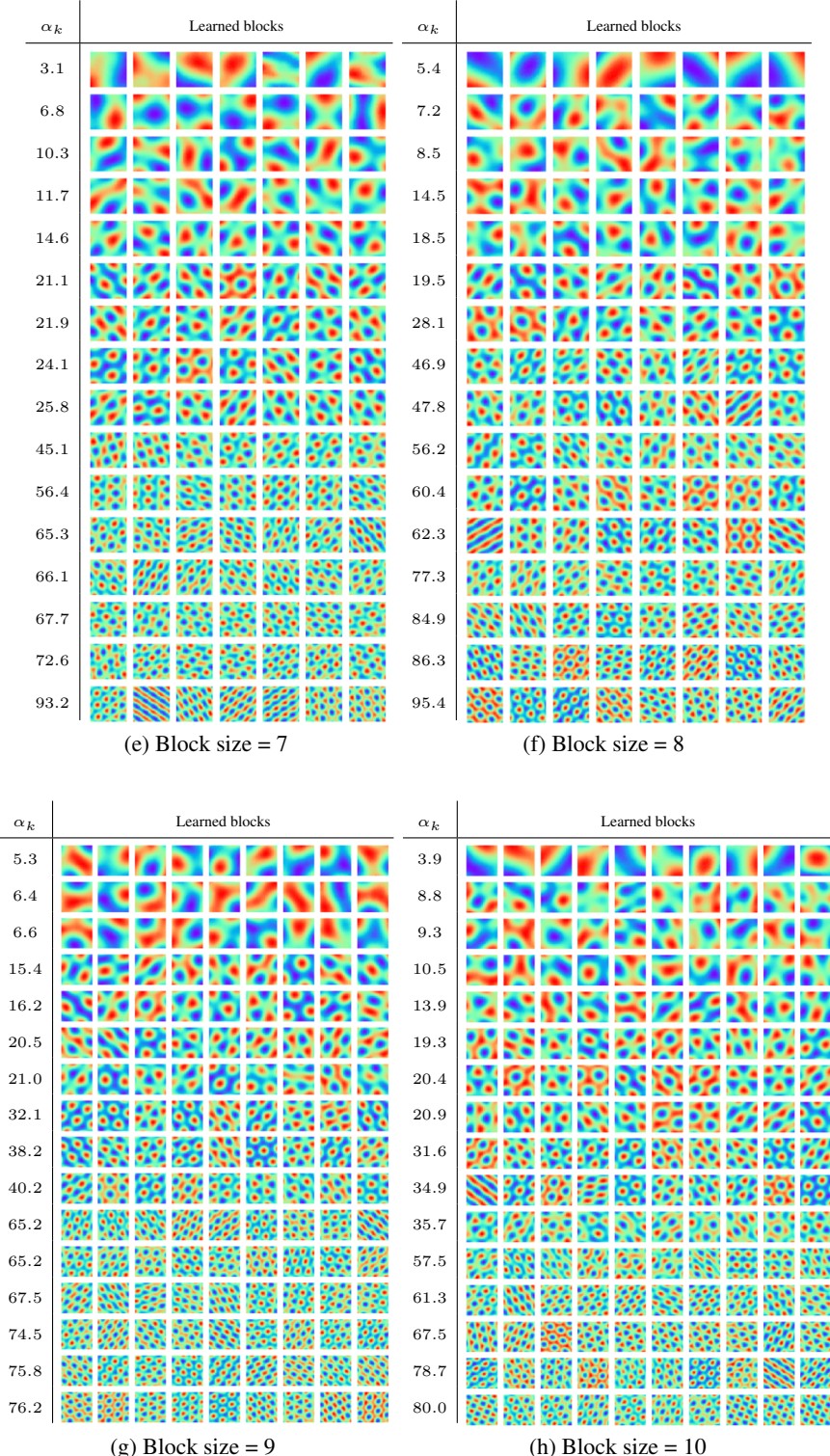

Figure 8: Learned multiple block units and metrics with different block sizes

## C  MODELING EGOCENTRIC MOTION

The model can be generalized to handle egocentric motion that involves head direction.

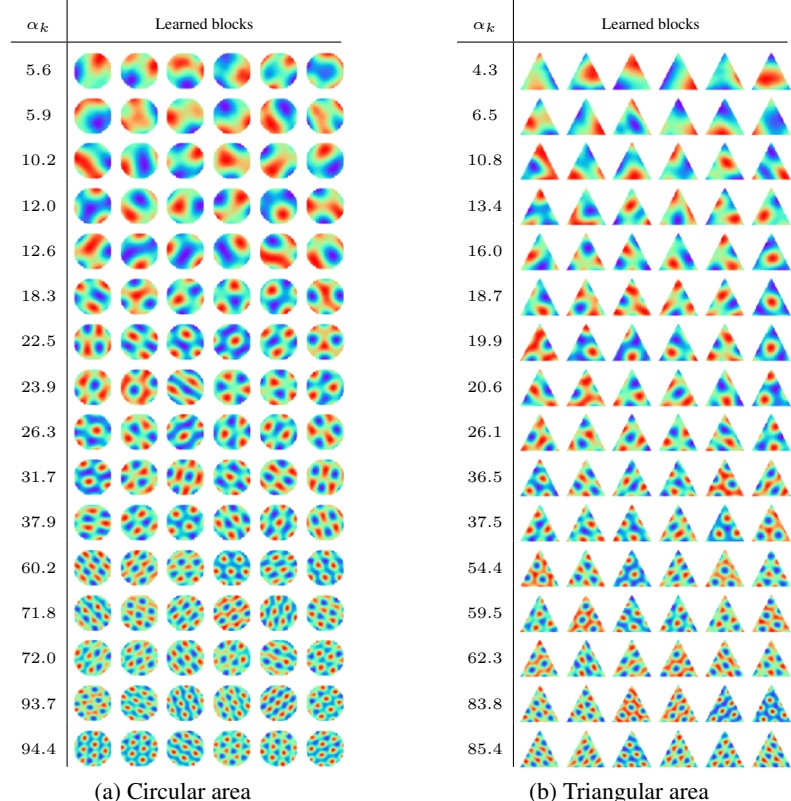

|  |  |
|---|---|
| (a) Circular area | (b) Triangular area |

Figure 9: Learned multiple block units and metrics with different shapes of the area

## C.1 COUPLING TWO GRID SYSTEMS

The egocentric motion consists of angular velocity in the change of head direction and the spatial velocity along the current head direction. We couple two grid systems, one for head direction and the other for the spatial position. For notational simplicity, we put the "hat" notation on top of $v$ and $M$ notation to denote the vector and matrix for the head direction.

Specifically, the agent has a head direction $\theta$, which may change over time. The agent moves along its head direction with scalar motion. We can discretize the range of $\theta$, $[0, 2\pi]$, into $\hat{N}$ equally spaced values $\{\theta_i, i = 1, ..., \hat{N}\}$, and introduce a vector representation of self-direction. That is, the agent represents its self-direction by a $\hat{d}$-dimensional hidden vector $\hat{v}(\theta)$.

### C.1.1 MOTION SUB-MODEL

Suppose at a position $x$, the agent has a head direction $\theta$. The self-motion is decomposed into (1) a scalar motion $\delta$ along the head direction $\theta$ and then (2) a head direction rotation $\Delta\theta$. The self-motion $\Delta x = (\delta \cos\theta, \delta \sin\theta)$. We assume

$$
\begin{aligned}
\hat{v}(\theta + \Delta\theta) &= \hat{M}(\Delta\theta)\hat{v}(\theta), & (31)\\
v(x + \Delta x) &= M(\delta, \hat{v}(\theta))v(x), & (32)
\end{aligned}
$$

where $\hat{M}(\Delta\theta)$ is a $\hat{d} \times \hat{d}$ matrix that depends on $\Delta\theta$, and $M(\Delta\theta, \hat{v}(\theta))$ is a $d \times d$ matrix that depends on $\delta$ and $\hat{v}(\theta)$. $\hat{M}(\Delta\theta)$ is the matrix representation of the head direction rotation $\Delta\theta$, while $M(\delta, \hat{v}(\theta))$ is the matrix representation of scalar motion $\delta$ along the direction $\theta$.

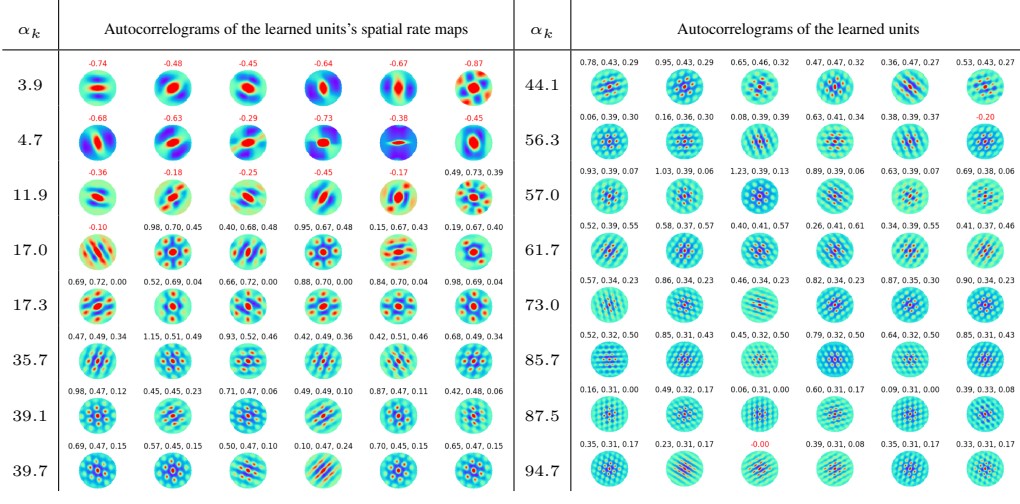

(a) Autocorrelograms of the learned units' response maps.

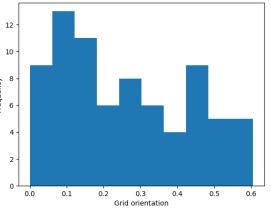
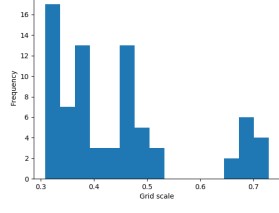
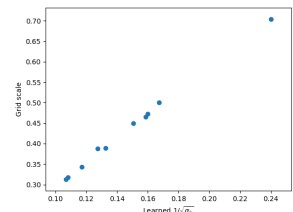

(b) Histogram of grid orientations

(c) Histogram of grid scales

(d) Scatter plot of grid scales and learned $1/\sqrt{\alpha_k}$

Figure 10: (a) Autocorrelograms of the learned units' response maps. Gridness scores are calculated based on the autocorrelograms. A unit is classified as a grid cell if the gridness score is larger than $0$. The gridness score is shown in red color if a unit fails to be classified as a grid cell. For those units that are classified as grid cells, gridness score, scale and orientation are listed sequentially in black color. Orientation is computed using a camera-fixed reference line ($0°$) and in counterclockwise direction. (b) Histogram of grid orientations. (c) Histogram of grid scales. (d) Scatter plot of averaged grid scales within each block versus the corresponding learned $1/\sqrt{\alpha_k}$.

We can model $M(\delta, \hat{v}(\theta))$ by an attention (or selection) mechanism:

$$p_i = \frac{\langle \hat{v}(\theta), \hat{v}(\theta_i) \rangle^b}{\sum_{i=1}^{\hat{N}} \langle \hat{v}(\theta), \hat{v}(\theta_i) \rangle^b}, \tag{33}$$

$$M(\delta, \hat{v}(\theta)) = \sum_{i=1}^{\hat{N}} p_i M^{(i)}(\delta), \tag{34}$$

where $M^{(i)}(\delta)$ is the matrix representation of scalar motion $\delta$ given the head direction $\theta_i$. The inner product $\langle \hat{v}(\theta), \hat{v}(\theta_i) \rangle$, that informs the angular distance between $\theta$ and $\theta_i$, serves as the attention weight. $b$ is an annealing (inverse temperature) parameter. If $b \to \infty$, $p = (p_i, i = 1, ..., \hat{N})$ becomes a one-hot vector for selection. We can further assume that $\hat{M}(\Delta\theta)$ and $M^{(i)}(\delta)$ are block diagonal, and learn a parametric model for each of them by the second order Taylor expansion at $\Delta\theta$ and $\delta$ respectively.

### C.1.2 LOCALIZATION SUB-MODEL

For the localization sub-model, we define the adjacency measures of self-direction and self-position separately. Let $\hat{f}(|\theta_1 - \theta_2|)$ be the adjacency measure between two directions $\theta_1$ and $\theta_2$. We use von Mises kernel $\hat{f}(r) = \exp((\cos(r) - 1)/\sigma^2)$, where $\hat{f}(0) = 1$. For adjacency measure between self-positions $x$ and $y$, we keep it the same as described in section 3.7.

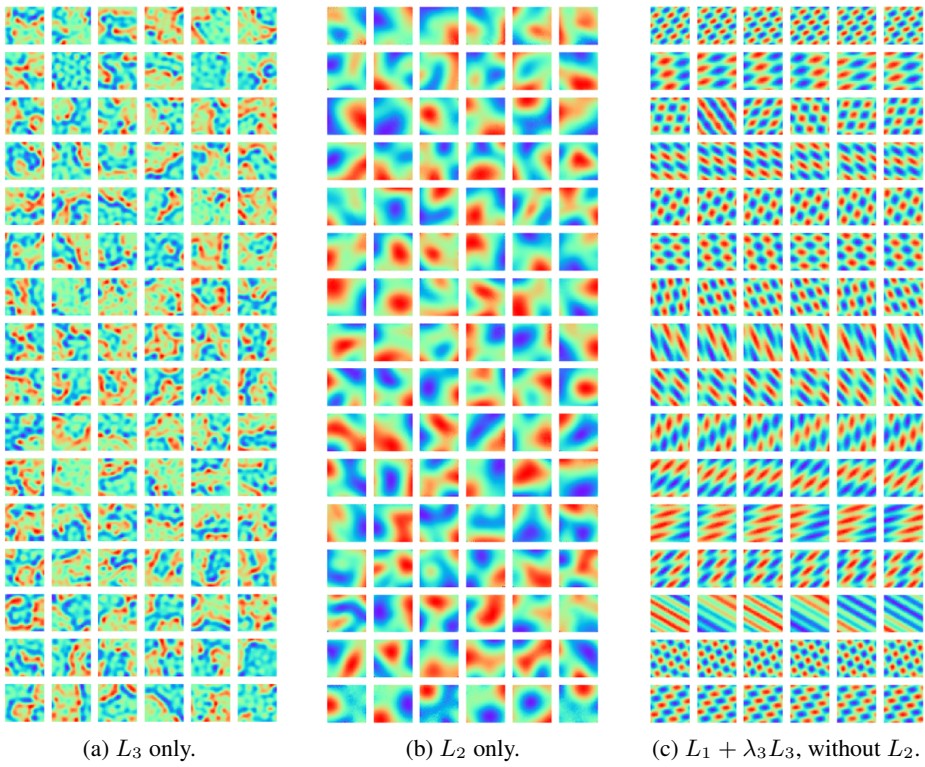

(a) $L_3$ only.                (b) $L_2$ only.                (c) $L_1 + \lambda_3 L_3$, without $L_2$.

Figure 11: Ablation study of the components in the training loss. (a) Learn the model using only the localization loss with global adjacency. (b) Learn the model using only the localization loss with local adjacency. (c) Using global adjacency and motion loss, leaving out local adjacency.

### C.1.3 LOSS FUNCTION FOR LEARNING

For learning the system, we can first learn $\hat{v}$ and $\hat{M}$ (or the coefficients that parametrize $\hat{M}$) by minimizing

$$\mathbb{E}_{\theta_1,\theta_2} \left[ (\hat{f}(|\theta_1 - \theta_2|) - \langle \hat{v}(\theta_1), \hat{v}(\theta_2) \rangle)^2 \right] + \lambda \mathbb{E}_{\theta,\Delta\theta} \left[ \| \hat{v}(\theta + \Delta\theta) - \hat{M}(\Delta\theta)\hat{v}(\theta) \|^2 \right], \qquad (35)$$

and then we learn $v$ and $M$ (or the coefficients that parametrize $M$) as before.

### C.2 LEARNED UNITS FOR SELF-DIRECTION AND SELF-POSITION

Figure 13 shows a result of learning such an egocentric motion model by displaying the response curves of the learned units in the head direction system, $\hat{v}(\theta)$, for $\theta \in [0, 2\pi]$, as well as the response maps of the learned multiple block units in the self-position system, $v(x)$, for $x \in [0, 1]^2$.

### C.3 CLOCK AND TIMESTAMP

We may re-purpose the head direction system as a clock, by interpreting $\theta \in [0, 2\pi]$ as the time on a clock, and $\hat{v}(\theta)$ as a timestamp for events happening over time. This may be related to the recent neuroscience observations in Tsao et al. (2018).

## D ERRORS

### D.1 ERROR CORRECTION

Unlike commonly used embedding in machine learning, here we embed a 2D position into a high-dimensional space, and the embedding is a highly distributed representation or population code. The

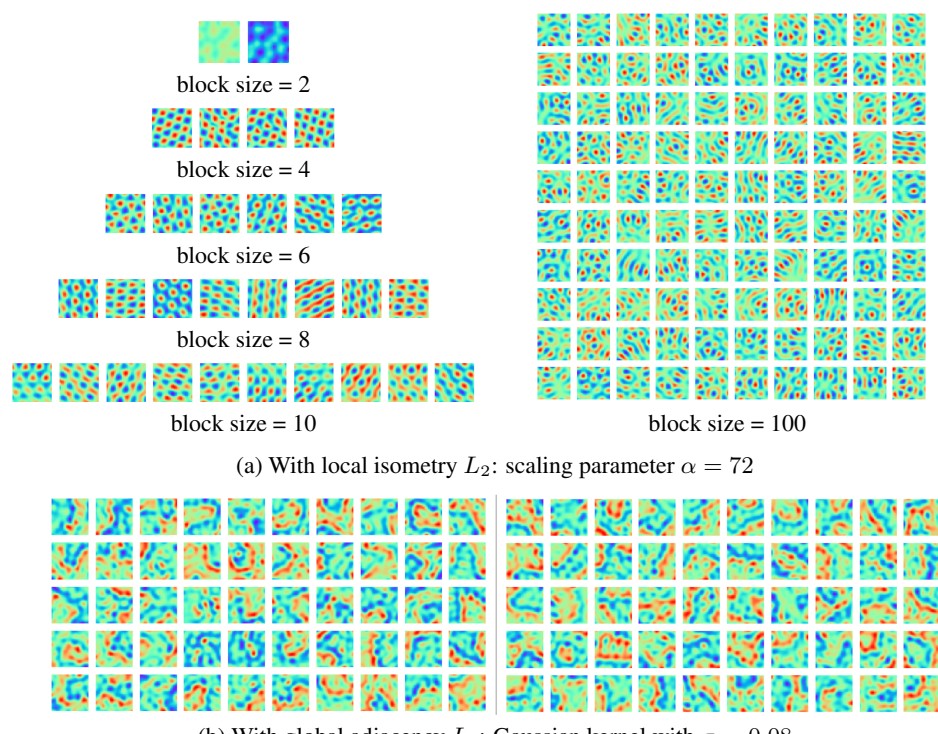

block size = 2

block size = 4

block size = 6

block size = 8

block size = 10

block size = 100

(a) With local isometry $L_2$: scaling parameter $\alpha = 72$

(b) With global adjacency $L_3$: Gaussian kernel with $\sigma = 0.08$

Figure 12: Learned units by dropping the parametrization and the block diagonal assumption of the motion matrix $M(\Delta x)$.

advantage of such a redundant code lies in its tolerance to errors. We show that the learned system is tolerant to various sources of errors. Specifically, in both path integral and path planning tasks, at every time step $t$, we randomly add (1) Gaussian noises or (2) dropout masks to the hidden units and see if the system can still perform the tasks well. We find that the decoding-encoding process (DE) is important for error correction. That is, at each time step $t$, given the noisy hidden vector $v_t$, we decode it to $x_t = \arg\max_x \langle v_t, v(x) \rangle$ and then re-encode it to the hidden vector $v(x_t)$. Actually the whole process can be accomplished by projection to the codebook sub-manifold without explicit decoding, by obtaining the vector: $\arg\max_{v \in C} \langle v_t, v \rangle$, where $C = \{v(x), x \in [0,1]^2\}$.

Table 1 shows the error correction results tested on path integral and path planning tasks. Each number is averaged over $1,000$ episodes. We compute the overall standard deviation of $\{v(x)\}$ for all $x$ and treat it as the reference standard deviation ($s$) for the Gaussian noise. For dropout noise, we set a percentage to drop at each time step. With the decoding-encoding process, the system is quite robust to the Gaussian noise and dropout error, and the system still works even if $70\%$ units are silenced at each step.

### D.2 NOISY SELF-MOTION INPUT

Besides adding noises to the hidden units, we also experiment with adding noises to the self-motion $\Delta x$, and compare the performance of path integral with the one performed in the original 2D coordinates. Specifically, at each time step, we add Gaussian noises to the self-motion $\Delta x$. For path integral, we compute the mean square error between the predicted locations and ground truth locations. Besides, we also compute the predicted locations using the 2D coordinates with the noisy self-motions. Its mean square error serves as the reference error. Table 2 shows the result, indicating that the error of the learned system is close to the error in the original 2D coordinates, i.e., our system does not blow up the noise.

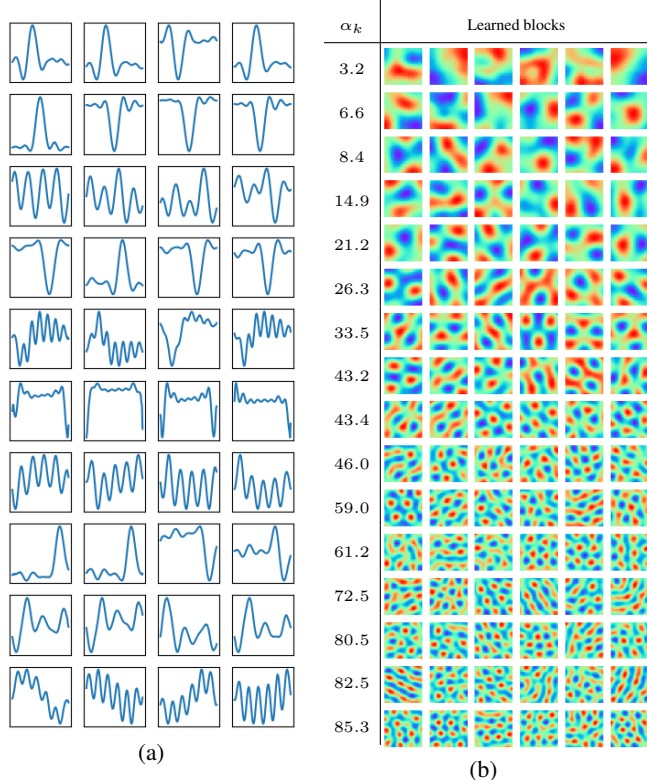

(a)    (b)

Figure 13: (a) Response curves of learned units in the head direction model, $\hat{v}(\theta)$. Each block shows the units belonging to the same sub-vector in the model. The horizontal axis represents the angle within a range of [0, $2\pi$], while the vertical axis indicates the values of responses. (b) Response maps of the learned multiple block units of the self-position model, $v(x)$, for $x \in [0,1]^2$.

| | Path integral: MSE | | | | | Path planning: success rate | | | | |
|---|---|---|---|---|---|---|---|---|---|---|
| Noise type | $1s$ | $0.75s$ | $0.5s$ | $0.25s$ | $0.1s$ | $1s$ | $0.75s$ | $0.5s$ | $0.25s$ | $0.1s$ |
| Gaussian (DE) | 1.687 | 1.135 | 0.384 | 0.017 | 0 | 0.928 | 0.959 | 0.961 | 0.977 | 0.985 |
| Gaussian | 6.578 | 2.999 | 1.603 | 0.549 | 0.250 | 0.503 | 0.791 | 0.934 | 0.966 | 0.982 |
| Noise type | 70% | 50% | 30% | 10% | 5% | 70% | 50% | 30% | 10% | 5% |
| Dropout (DE) | 2.837 | 1.920 | 1.102 | 0.109 | 0.013 | 0.810 | 0.916 | 0.961 | 0.970 | 0.978 |
| Dropout | 19.611 | 16.883 | 14.137 | 3.416 | 0.602 | 0.067 | 0.186 | 0.603 | 0.952 | 0.964 |

Table 1: Error correction results on the vector representation. The performance of path integral is measured by mean square error between predicted locations and ground truth locations; while for path planning, the performance is measured by success rate. Experiments are conducted using several noise levels: Gaussian noise with different standard deviations in terms of the reference standard deviation $s$ and dropout mask with different percentages. DE means implementing decoding-encoding process when performing the tasks.

## E    3D ENVIRONMENT AND 1D TIME

The system can be generalized to 3D environments. Specifically, we assume the agent navigates within a domain $D = [0,1] \times [0,1] \times [0,1]$, which is discretized into a $40 \times 40 \times 40$ lattice. We learn a parametric model for motion matrix $M$ in a residual form $M = I + \tilde{M}(\Delta x)$, where $\tilde{M}(\Delta x)$ is approximated by the second order Taylor expansion of $\Delta x$.

We first learn a single block with fixed $\alpha_k$ by minimizing $L_1 + \lambda_2 L_{2,k}$. A batch of 500,000 examples of $(x, y)$ and $(x, \Delta x_t, t = 1, ..., T)$ is sampled online at every iteration for training. Figure 14 shows the learned units.

| Standard deviation | 1.2 | 0.9 | 0.6 | 0.3 |
|---|---|---|---|---|
| Learned system | 6.020 | 4.382 | 3.000 | 1.422 |
| Reference | 5.852 | 4.185 | 2.873 | 1.315 |

Table 2: Path integral results with noises in the self-motion. Performance is measured by mean square error (MSE) between the predicted locations and ground truth locations in the path integral task. Noises are added to self-motions $\Delta x$ by several noise levels: Gaussian noise with different standard deviations in terms of number of grids. Reference MSE is computed by path integral in 2D coordinates.

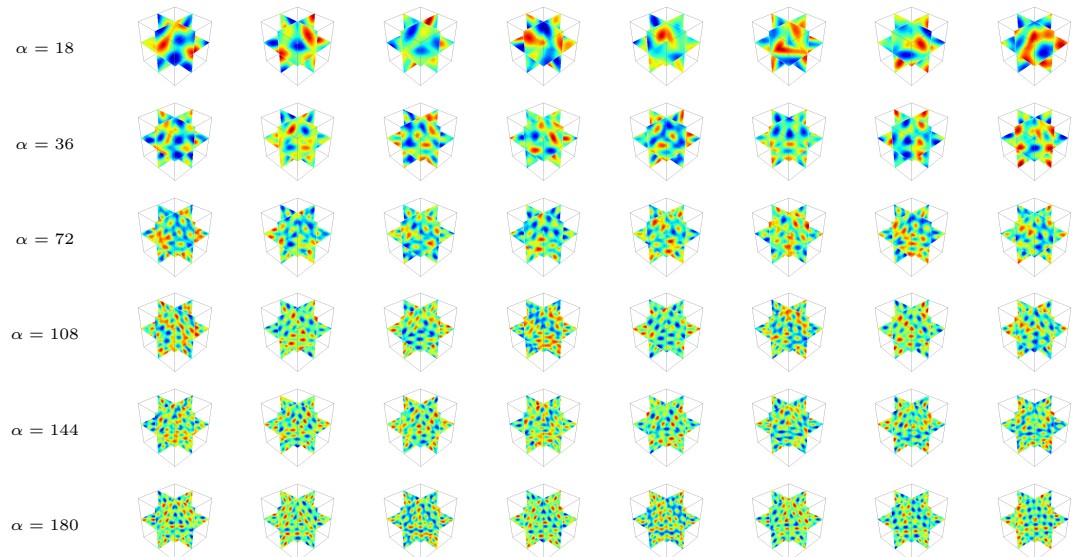

Figure 14: Learned units of a single block with fixed $\alpha$ in 3D environment. Every row shows the learned units with a given $\alpha$.

Next we learn multiple blocks by minimizing $L_1 + \lambda_3 L_3$, and use the learned models to perform 3D path integral and path planning. For simplicity, we remove $L_2$. We use 8 blocks of units with block size 8 and exponential kernel ($\sigma = 0.3$) for path planning. A batch of 200,000 examples is sampled online at every iteration for training.

## E.1 3D PATH INTEGRAL

Figure 15 shows some results of 3D path integral with duration $\tilde{T} = 30$. Gaussian Kernel ($\sigma = 0.08$) is used as the adjacency measure.

## E.2 3D PATH PLANNING

Figure 16 shows some results of 3D simple path planning. Exponential kernel ($\sigma = 0.3$) is used as the adjacency measure. We design a set of allowable motions $\Delta$: $m$ lengths of radius $r$ are used, and for each $r$, we evenly divide the inclination $\theta \in [0, \pi]$ and the arimuth $\alpha \in [0, 2\pi]$ into $n$ directions, which results in a motion pool with $mn^2$ candidate motions $\Delta = \{(r\sin(\theta)\cos(\alpha), r\sin(\theta)\sin(\alpha), r\cos(\theta))\}$. We use $m = 2, n = 90$ in this experiment.

## E.3 3D PATH PLANNING WITH OBSTACLES

Figure 17 shows some examples of 3D path planning with a cuboid obstacle. $a = 38$ and $b = 24$ in equation 16.

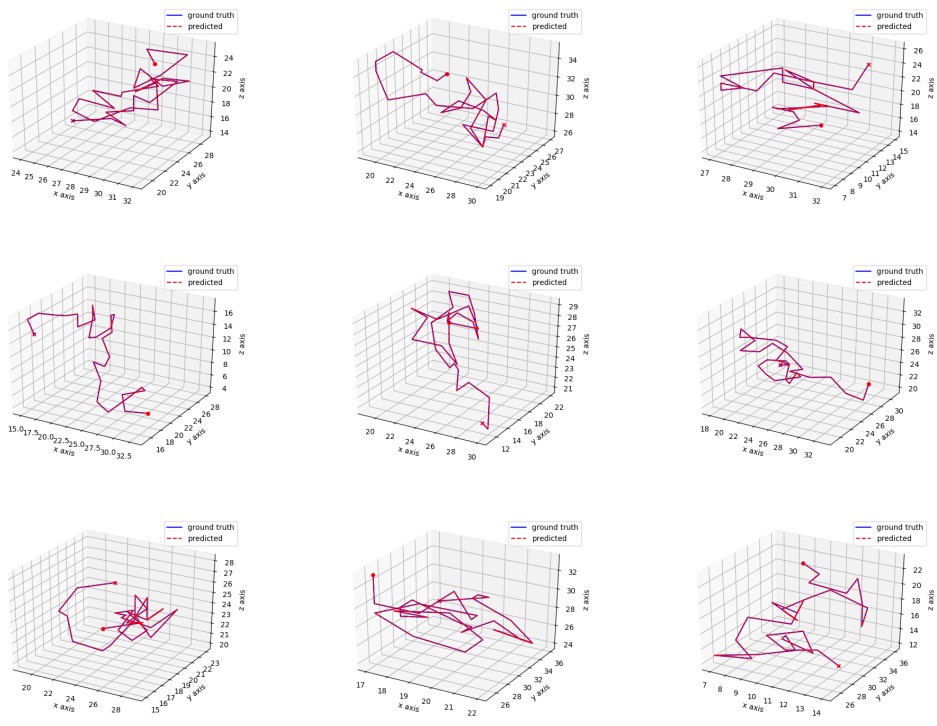

Figure 15: Examples of 3D path integral with duration $\tilde{T} = 30$.

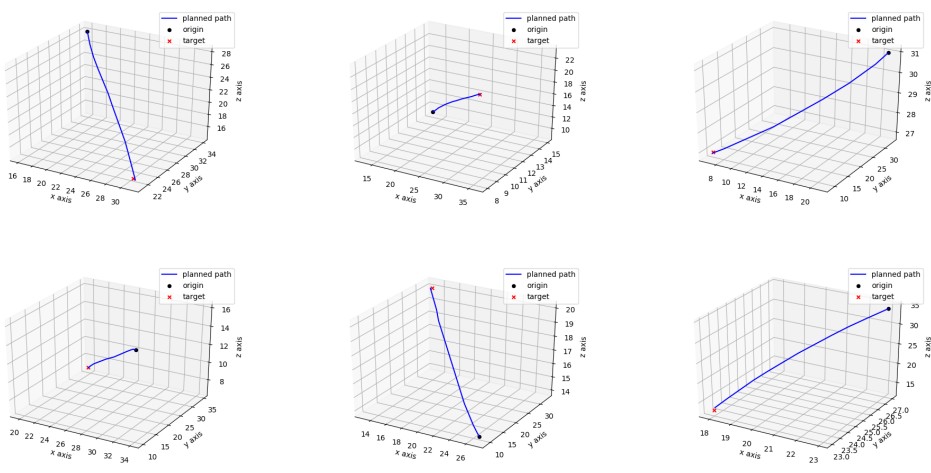

Figure 16: Examples of 3D simple path planning, where the agent is capable of planning a direct trajectory.

### E.4 LEARNING IN 1D

The system can also be applied to 1D. Inspired by Tsao et al. (2018), the learned system in 1D may serve as a timestamp of events. We assume domain $D = [0, 1]$ and discretize it into 100 time points. A parametric $M$ is learned by a residual form $M = I + \tilde{M}(\Delta t)$, where each element of $\tilde{M}(\Delta t)$ is parametrized as a function of $(\Delta t, \Delta t^2)$. 16 blocks of hidden units with block size 6 are used. Figure 18 visualizes the learned units over the 100 time points. The response wave of every unit

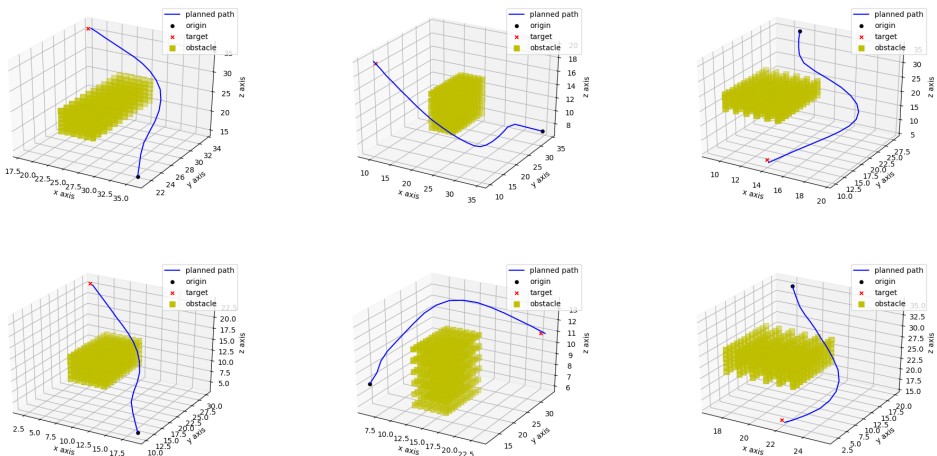

Figure 17: Examples of 3D path planning with a cuboid obstacle.

shows strong periodicity of a specific scale or frequency. Within each block, the response waves of units have similar patterns, with different phases.

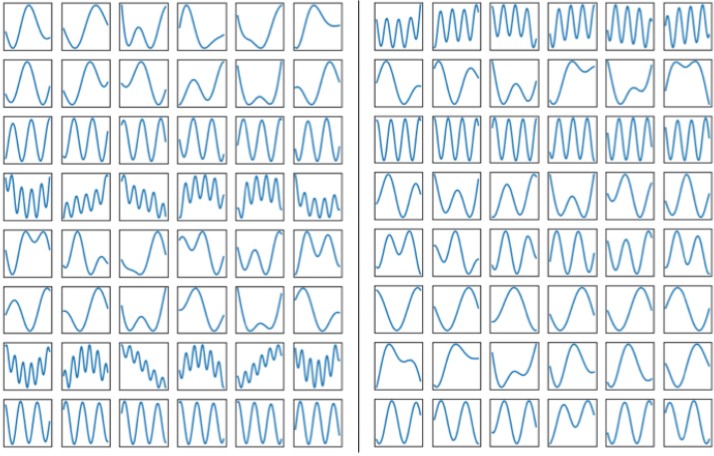

Figure 18: Response curves of learned units in 1D. Each block shows the units belonging to the same sub-vector in the motion model. The horizontal axis represents the time points, while the vertical axis indicates the value of responses.

