# OpenReview forum: "Learning Grid Cells as Vector Representation of Self-Position Coupled with Matrix Representation of Self-Motion"
_ICLR.cc/2019/Conference_

### Official Review · AnonReviewer2 · 2018-10-31
**The motivation for this work needs to be clarified**

**Rating:** 7
**Confidence:** 4

**Review:**



=Major Comments=
The prior work on grid cells and deep learning makes it clear that the goal of the work is to demonstrate that a simple learning system equipped with representation learning will produce spatial representations that are grid-like. Finding grid-like representations is important because these representations occur in the mammalian brain.

Your paper would be improved by making a similar argument, where you would need to draw much more explicitly on the neuroscience literature. Namely, the validation of your proposed representations for position and velocity are mostly validated by the fact that they yield grid-like representations, not that they are useful for downstream tasks.

Furthermore, you should better justify why your simple model is better than prior work? What does the simplicity provide? Interpretability? Ease if optimization? Sample complexity for training?

This is important because otherwise it is unclear why you need to perform representation learning. The tasks you present (path integral and planning) could be easily performed in basic x-y coordinates. You wouldn’t need to introduce a latent v. Furthermore, this would mprove your argument for the importance of the block-diagonal M, since it would be more clear why interpretability matters.


Finally, you definitely need to discuss the literature on randomized approximations to RBF kernels (random Fourier features). Given the way you pose the representation learning objective, I expect that these would be optimal. With this, it is clear why grid-like patterns would emerge.

=Additional Comments=
What can you say about the quality of the path returned by (10)? Is it guaranteed to converge to a path that ends at y? Is it the globally optimal path?

I don’t agree with your statement that your approach enables simple planning by steepest descent. First of all, are the plans that your method outputs high-quality? Second, if you had solved (10) directly in x-y coordinates, you could have done this easily since it is an optimization problem in just 2 variables. That could be approximately solved by grid search.

I would remove section 5.4. The latent vector v is a high-dimensional encoding of low-dimensional data, so of-course it is robust to corruptions. The corruptions you consider don’t come from a meaningful noise process, however? I can imagine, for example, that the agent observes corrupted versions of (x,y), but why would v get corrupted?

---

> ### Author Response · Authors · 2018-11-26
> **Reply to Reviewer 2 (part 1)**
>
> Thank you for your helpful comments and suggestions.
>
> Q1:“Your paper would be improved by making a similar argument, where you would need to draw much more explicitly on the neuroscience literature.”
>
> A1: Your advice is followed. We have added a discussion in the related work (Section 2), indicating that the disentangled blocks assumption in our model is related to the “modules” of grid cells. We have also added a quantitative analysis using measures from the neuroscience literature to analyze the spatial activity of the learned units. Please see Section B.3 of the appendix.
>
>
> Q2: “Furthermore, you should better justify why your simple model is better than prior work? What does the simplicity provide? Interpretability? Ease if optimization? Sample complexity for training? This is important because otherwise it is unclear why you need to perform representation learning. The tasks you present (path integral and planning) could be easily performed in basic x-y coordinates. You wouldn’t need to introduce a latent v. Furthermore, this would improve your argument for the importance of the block-diagonal M, since it would be more clear why interpretability matters.”
>
> A2: Thanks for the thoughtful comments, which we agree.
>
> The simplicity here is about explaining the patterns observed in grid cells, and simplicity is desired or even required of an explanation of an observed phenomenon.
>
> In particular, in Section 5.1 of the revised version, we show that the emergence of the global hexagon patterns can be explained by a generic local kernel and a generic local motion model, both of which are very simple.
>
> In our work, we show that this simple system is capable of path integral and path planning.
>
> We agree with you that these tasks can be performed in the 2D coordinates. It is a deep question as to why the mammalian brain uses a latent v. The justification we can provide is that the system with a high-dimensional v is capable of error correction, considering the neural system is intrinsically noisy. But there may be deeper or stronger justifications. One speculation is that the neural system may prefer matrix-vector multiplication to addition and subtraction.
>
>
> Q3:“Finally, you definitely need to discuss the literature on randomized approximations to RBF kernels (random Fourier features). Given the way you pose the representation learning objective, I expect that these would be optimal. With this, it is clear why grid-like patterns would emerge.”
>
> A3:  Thanks for the reference and the insight. We have cited the related papers and compare them to our work at the end of Section 5.2.
>
> In Section 5.1 of the revised version, we show that a local radial basis kernel and a local motion model are enough to explain the emergence of the global hexagon patterns of the grid cells. In Appendix A, we also provide a theoretical understanding.
>
> Inspired by your comment, we have added an ablation study in Section B.4.1 of the appendix, we show that the motion model v(x+dx) = M(dx) v(x) is necessary for the emergence of the grid patterns. We cannot learn the grid patterns from the localization model A(x, y) = <v(x), v(y)> alone.
>
> Compared to random Fourier features, we learn the grid patterns without assuming Fourier basis, our RBF kernel is a local generic one based on the second order Taylor expansion, and we need a motion model for the emergence of grid patterns.

---

> > ### Author Response · Authors · 2018-11-26
> > **Reply to Reviewer 2 (part 2)**
> >
> > Q4:“What can you say about the quality of the path returned by (10)? Is it guaranteed to converge to a path that ends at y? Is it the globally optimal path? I don’t agree with your statement that your approach enables simple planning by steepest descent. First of all, are the plans that your method outputs high-quality? Second, if you had solved (10) directly in x-y coordinates, you could have done this easily since it is an optimization problem in just 2 variables. That could be approximately solved by grid search. ”
> >
> > A4: The convergence of a path can be quantified by success rate. Specifically, we can define a path planning to be successful if the distance between the agent’s end position and the target is less than 0.025, and the distance between each point on the path and the obstacle is larger than 0.025. When a = 0.5 and b = 6, the successful rate is larger than 99%.
> >
> > We agree with your criticism of our statement. We have removed the statement that compares our method with reinforcement learning and path planning. We have re-positioned our work on path planning, only claiming that our system is capable of implementing a path planning algorithm that is similar to the potential field method in robotics, thus sharing the advantages and disadvantages of the latter. Please see the paragraph at the beginning of Section 5.4.
> >
> > We agree with your comment about solving path planning in 2D coordinates. Our claim is only that our system based on (v(x), M(dx), A(x, y)) is capable of implementing path planning algorithms based on (x, dx, |x-y|) (here both x and y are 2D). This is actually non-trivial. A learned grid cells system that is capable of path integral is not necessarily capable of path planning. The fact that A(x, y) informs |x-y| in our system is important for path planning.
> >
> > As to why the mammalian brain adopts the grid cells instead of directly representing the 2D coordinates (e.g., by two neurons), our explanation is that the high-dimensional v enables error correction.
> >
> >
> > Q5: “I would remove section 5.4. The latent vector v is a high-dimensional encoding of low-dimensional data, so of-course it is robust to corruptions. The corruptions you consider don’t come from a meaningful noise process, however? I can imagine, for example, that the agent observes corrupted versions of (x,y), but why would v get corrupted?”
> >
> > A5: Following your advice, we have moved the error correction part to the appendix. See Section D of the appendix.
> >
> > The units in v are neurons, and they tend to be noisy in the biological system. The dropout may also be related to the asynchronous nature of neuron activities. Dropout may also be caused by the gradual loss of neurons due to aging or Alzheimer.
> >
> > Error correction may provide a justification for the high-dimensional vector encoding of the two-dimensional coordinates.

---

> > > ### Comment · AnonReviewer2 · 2018-11-26
> > > **Thank for such a thorough response**
> > >
> > > You have addressed my questions very well, and I appreciate that you have updated the document so much. I have raised my evaluation score.

---

### Official Review · AnonReviewer3 · 2018-11-02
**Elegant but simplistic model for grid cells; unnecessary extension to path planning**

**Rating:** 7
**Confidence:** 5

**Review:**

Updated score from 6 to 7 after the authors addressed my comments below.

Previous review:

This paper builds upon the recent work on computational models of grid cells that rely on trainable (parametric) models such as recurrent neural networks [Banino et al, 2018; Cueva & Wei, 2018]. It focuses entirely on path integration in 2D and 3D, from velocity inputs only, and it relies on two sub-networks: the motion model (an RNN) and the localization model (a feed-forward network). The avowed goal of the paper is to build a very simple and linear model for grid cells.

By linearly embedding the position x into a high-dimensional hidden vector v(x) (e.g., 96 elements), it can model motion using a linear model relying on matrix-vector multiplication: v(x + dx) = M(dx) v(x), where dx is the 2D or 3D displacement, v(.) is a vector and M(.) is a matrix. The embeddings v(.) are learnable and the paper assumes a square or cubic grid of N*N or N*N*N possible positions x (with N=40); these embeddings are also normalized to unit length and obey the kernel constraint that the dot-product between any two positions' vectors v(x) and v(y) is a Gaussian or an exponential function. The motion matrix is represented as block diagonal, where each block is a rotation of subvector v_k(x) into v_k(x + dx), where each block corresponds to a specific grid cell, and where the diagonal block is further expressed as a quadratic function of dx_1, dx_2, dx_3 elements of the displacement vector.

The strengths of the paper are that:
1) The supervision of the localization subnetwork only depends on Euclidean proximity between two positions x and y and therefore uses relative positions, not absolute ones. Similarly, the path integration supervision of the motion model uses only relative displacements.
2) The resulting rate maps of the hidden units seem perfect; the model exhibits multi-scale grid behaviour.
3) The idea of using disentangled blocks, rather than injecting noise or using dropout and a softmax bottleneck as in [Banino et al, 2018], is interesting.
4) The model accumulates little path integration error over 1000 step-long episodes.

The weakness of the paper is its simplicity:
1) The assumption that A(x, y) can be modeled by a Gaussian or exponential (Laplacian?) kernel is limiting, in particular for positions x and y that are far apart.
2) There is no discussion about what egocentric vs. allocentric referentials, and dx is assumed to be aligned with (x, y) axes (which are also the axes defining the bounding box of the area).
3) Unlike the other work on learning path integration using an RNN, the linear matrix model can only handle allocentric displacements dx_1, dx_2 (and optional dx_3 in 3D).
4) No consideration is given to non-square areas: would the network also exhibit grid-like behavior if the area was circular?
5) What happens if the quadratic parameterisation of block diagonals is dropped?
6) The paper did not use metrics accepted in the neuroscience community for computing a gridness score of the grid cells (although the grid cell nature is evident). There should however be metrics for quantifying how many units represent the different scales, offsets and orientations.
7) The authors did not consider (but mentioned) embedding locations from vision, and did not consider ambiguous position embeddings.

The experiments about path planning are unconvincing. First of all, the algorithm requires to input absolute positions of every obstacle into equation (9) - (10), which assumes that there is perfect information about the map. Secondly, the search algorithm is greedy and it is not obvious how it would handle a complex maze with cul-de-sac. Saying that "there is no need for reinforcement learning or sophisticated optimal control" is very misleading: the problem here is simplified to the extreme, and fully observed, and any comparison with deep RL algorithms that can handle partial observations is just out of place.

In summary, the authors have introduced an interesting and elegant model for grid cells that suffers from simplifications. The part on path planning should be cut and replaced with more analysis of the grid cells and an explanation of how the model would handle egocentric velocity.

---

> ### Author Response · Authors · 2018-11-26
> **Reply to Reviewer 3 (part 1)**
>
> Thank you for the helpful comments and suggestions.
>
>
> Q1: “The assumption that A(x, y) can be modeled by a Gaussian or exponential (Laplacian?) kernel is limiting, in particular for positions x and y that are far apart.”
>
> A1: We agree with your concern with the global adjacency. We have studied a generic local adjacency based on the second order Taylor expansion. Please see Sections 5.1, 5.2, and Section B of the appendix.
>
> This generic local adjacency 1 – alpha |x-y|^2 appears to be the key for the emergence of the global hexagon grid pattern where alpha determines the metric.
>
> Meanwhile, the global adjacency is necessary for the following two reasons. (1) Regulate the metrics of multiple blocks of the hexagon grid units. (2) Inform |x-y| for the purpose of path planning.
>
>
> Q2: “There is no discussion about what egocentric vs. allocentric referentials, and dx is assumed to be aligned with (x, y) axes (which are also the axes defining the bounding box of the area). Unlike the other work on learning path integration using an RNN, the linear matrix model can only handle allocentric displacements dx_1, dx_2 (and optional dx_3 in 3D).”
>
> A2: Inspired by your comment, we have added a section on egocentric model. Please see Section C of the appendix.
>
> The model couples the grid system for head direction and the original grid system for self-position. The coupling is as follows: the vector of the head direction system determines the matrix of the self-position system via an attention or selection mechanism. We find this model quite interesting although we still need more work to refine it.
>
> The head direction system can also be repurposed as a clock and timestamp system.
>
>
> Q3: “No consideration is given to non-square areas: would the network also exhibit grid-like behavior if the area was circular?”
>
> A3: To answer your question, we learn the system in circular and triangular areas and the results are shown in Figure 7 of Section B.2.2 of the appendix. Hexagon patterns emerge in both cases.
>
>
> Q4: “What happens if the quadratic parametetrisation of block diagonals is dropped?”
>
> A4: To answer your question, we have added an ablation study in Section B.4.2 of the appendix, where we remove the block diagonal assumption and the quadratic parametrization, so that we learn a separate motion matrix for each displacement on the discretized 2D grid. With local adjacency, we can still learn hexagon grid patterns when the block size is relatively small. For global adjacency, we cannot learn hexagon grid patterns.
>
> Q5: “The paper did not use metrics accepted in the neuroscience community for computing a gridness score of the grid cells (although the grid cell nature is evident). There should however be metrics for quantifying how many units represent the different scales, offsets and orientations.”
>
> A5: Following your advice, we have added a quantitative analysis in Section B.3 of the appendix, using the measures from the neuroscience literature, including gridness score, grid scale and orientation. 76 out of 96 units are classified as grid units according the gridness score.
>
> An interesting result is that the scale measure is proportional to the metric (1/sqrt(alpha_k)) explicitly defined and automatically learned by our method. Please see Figure 8.d.
>
>
> Q6: “The authors did not consider (but mentioned) embedding locations from vision, and did not consider ambiguous position embeddings.”
>
> A6: To address your comment, we have added the following paragraph in Section 3.2 to discuss embedding location for vision.
>
>  “Our system can be embedded into the SLAM (simultaneous localization and mapping) system (\cite{whyte2006simultaneous}), which is based on a state space model that consists of a dynamic sub-model for self-position due to self-motion, and an observation sub-model for the observed visual image given the self-position. We can represent the dynamic sub-model by our system, or reformulate the whole model using our scheme. We leave it to future work. ”
>
> We are currently pursuing this direction of research.
>
> For ambiguous position embeddings, in Section D of the appendix, we consider errors in the units and show that our system is capable of error correction. We also added Section D.2 about noisy input of self-motion.

---

> > ### Author Response · Authors · 2018-11-26
> > **Reply to Reviewer 3 (part 2)**
> >
> > Q7: “The experiments about path planning are unconvincing. First of all, the algorithm requires to input absolute positions of every obstacle into equation (9) - (10), which assumes that there is perfect information about the map. Secondly, the search algorithm is greedy and it is not obvious how it would handle a complex maze with cul-de-sac. Saying that "there is no need for reinforcement learning or sophisticated optimal control" is very misleading: the problem here is simplified to the extreme, and fully observed, and any comparison with deep RL algorithms that can handle partial observations is just out of place.”
> >
> > A7: We agree with your criticism. We have removed the statement about reinforcement learning and optimal control. We have re-positioned our work on path planning, by only claiming that our system is capable of implementing a path planning algorithm that is similar to the potential field method in robotics, thus sharing the advantages and disadvantages of the latter. Please see the paragraph at the beginning of Section 5.4.
> >
> > Now the purpose of this section is only to show that: our (v(x), M(dx), A(x, y)) system is capable of implementing path planning algorithms based on (v, dx, |x-y|), even though our system does not represent the 2D coordinates x = (x1, x2) explicitly. This is actually non-trivial. A learned system that is capable of path integral is not necessarily capable of path planning. The fact that A(x, y) informs |x-y| in our system is important for path planning.
> >
> > We suspect that we need both path planning algorithm and a learned policy. The latter may be useful in a familiar environment, while the former may be necessary in an unfamiliar environment. During the path planning process, the grid cells are expected to be active even though the agent is not moving.

---

> ### Comment · AnonReviewer3 · 2018-11-26
> **Thank you**
>
> The response is thorough and my concerns are addressed, I have updated the score accordingly.

---

### Official Review · AnonReviewer1 · 2018-11-06
**A simple and elegant approach to grid cells that begs for theoretical insight**

**Rating:** 8
**Confidence:** 4

**Review:**

This paper proposes a simple and elegant approach to learning "grid-cell like" representations that uses a high-dimensional encoding of position, together with a matrix for propagating position that involves only local connections among the elements of the vector.  The vectors are also constrained to have their inner products reflect positional similarity.  The paper also shows how such a representation may be used for path planning.

By stripping away the baggage and assumptions of previous approaches, I feel this paper starts to get at the essence of what drives the formation of grid cells.   It is still steps away from having direct ties to neurobiology, but is trying to get at the minimal components necessary for bringing about a grid cell like solution.  But I feel the paper also stops just a few steps short of developing a fuller theoretical understanding of what is going on.  For example the learned solution is quite Fourier like, and we know that Fourier transforms are good for representing position shift in terms of phase shift.  That would correspond to block size of two (i.e., complex numbers) in terms of this model.  So what's wrong with this solution (in terms of performance) and what is gained by having block size of six, beyond simply looking more grid like?  It would be nice to go beyond phenomenology and look at what the grid-like solution is useful for.

---

> ### Author Response · Authors · 2018-11-26
> **Reply to Reviewer 1**
>
> We are very grateful for your positive review and insightful comments.
>
>
> Q1: “But I feel the paper also stops just a few steps short of developing a fuller theoretical understanding of what is going on.”
>
> A1: Following your advice, we have added theoretical analysis. Please see Section 5.1.3 and Section A of the appendix.
>
> In the theoretical analysis, we provide an analytical solution that combines three Fourier plane waves. The analysis is based on a tight frame in 2D.
>
> We believe this analytical solution helps us understand the emergence of hexagon patterns. Meanwhile, our model assumes much less than the analytical solution.
>
>
> Q2: “For example the learned solution is quite Fourier like, and we know that Fourier transforms are good for representing position shift in terms of phase shift. That would correspond to block size of two (i.e., complex numbers) in terms of this model. So what's wrong with this solution (in terms of performance) and what is gained by having block size of six, beyond simply looking more grid like? It would be nice to go beyond phenomenology and look at what the grid-like solution is useful for.”
>
> A2:  Thanks for the insight.
>
> Restricting block size = 2 indeed enables us to learn Fourier plane waves. Please see Figure 7.a.
>
> Figure 3.c shows that the path integral error with block size 2 is bigger than other block sizes.
>
> In terms of localization sub-model, a single pair of Fourier plane waves v(x) = exp(i<a,x>) in a block gives us an adjacency function <v(x), v(y)> = cos(<a, x-y>), which does not inform |x-y| very well due to the aperture problem, i.e., if x-y is perpendicular to a, the adjacency is always 1.
>
> In our new result, if we assume a generic local kernel <v(x), v(y)> = 1 – alpha |x-y|^2, then we can always learn hexagon grid patterns as long as the block size is greater than or equal to 6, where alpha controls the metric of the block.

---

### Author Response · Authors · 2018-11-07
**Surprising new result: hexagon and metric, with theoretical analysis**

Dear Reviewers,

Thank you for your precious time and insightful comments.

We have uploaded the first revision to include a new result that we find surprising, interesting and important. Please see Subsection 5.1 of the first revision.

To summarize, we learn a single block of cells with a generic local kernel. Recall the adjacency A(x, y)=f(|x-y|). A second order Taylor expansion of f(r) at r = 0 gives us f(r) = 1-alpha r^2, for small r, where 2 alpha is the curvature of f(r) at 0. The first derivative is 0 because f(r) reaches maximum at 0. We use the localization loss term

|<v(x), v(y)> - (1-alpha |x-y|^2)|^2,  for |x-y|^2 <= 1.5/alpha.

Together with the motion loss, our learning method unfailingly produces hexagon patterns, and alpha determines the metric or grid size. The hexagon patterns emerge as long as the number of units is greater than or equals to 6 (for smaller number we may learn rectangle patterns).

We also provide a theoretical solution when the number of units equals to 6, based on a tight frame in 2D.

We want to emphasize that both the localization loss and the motion loss are LOCAL, and yet the global hexagon patterns always emerge. Our loss function does not assume any global periodic pattern. It is perhaps the simplest loss function one can find: (1) a second order Taylor expansion for localization loss. (2) a matrix-vector product for motion loss. This is really minimalistic. That is, what we put in is far less than what we get out.

We believe this is the most important result of our work, because after all, the grid cells are characterized by hexagon patterns of different sizes. This is why they are called grid cells in the first place. We now can explain this crucial piece of puzzle.

We will incorporate this local kernel loss term into our original global kernel loss so that we will learn the metric alpha for each block automatically.

To save space, we moved the 3D path planning to appendix. We also added 1D result in appendix. The 1D result can be interpreted as time2vec or time stamp for events.

A few key points:

We shall reply to your valuable comments soon and further revise our paper according to your comments and advice. But first please allow us to make a few key points here:

(1)	In our representation scheme, we NEVER represent the coordinates x = (x1, x2) explicitly. We only represent the position by heat map or one-hot map. Without explicit coordinates, it is not a trivial task to do path planning, and it is very different from path planning in robotics based on explicit coordinates.
(2)	About path planning. Consider a rat leaves his home to forage. He needs path integration to know where he is. But MORE importantly, when he needs to go back home, he needs path planning. Even when he is standing still, his grid cells are changing during path planning, i.e., he is imagining or fantasizing the steps. Our proposed steepest ascent algorithm is of this nature.  The rat can also fantasize much bigger step sizes beyond his physical capability in path planning, and our method enables him to do that, see Figure 4(a) for straight path planning.
(3)	About error correction. When talking to people in CS and robotics, a common question is: how come the brain does not use two neurons to represent the two coordinates x = (x1, x2), and instead use many neurons to represent the position. Our error correction experiment may give a justification. The dropout experiment also points to the possibility that the grid cells can work asynchronously, which is typical of biological neural system. We shall explore this issue further.

We shall reply to your comments and upload the second revision soon.

Thank you for your consideration of our first revision and first reply.

---

### Author Response · Authors · 2018-11-10
**Surprising new result (continued): multiple hexagon blocks with automatically learned metrics**

Dear Reviewers,

We have uploaded the second revision that includes a new Subsection 5.2 on learning multiple blocks of grid cells where the metrics or grid sizes alpha_k are automatically learned.

Figure 2.a shows the learned blocks and their learned metrics alpha_k. You can see that the learned blocks again show hexagon patterns, and different blocks have different metrics or grid sizes. The metrics are explicitly defined as the curvatures of the local kernels and are learned together with the vector and matrix representations.

In Figure 2.a, the number of cells in each block is 6. In Appendix D, we show the learned blocks with different numbers of cells. As long as the number is greater than or equal to 6, the hexagon patterns emerge (for smaller number, the learned cells tend to exhibit square lattice patterns).

Figure 2.b shows the heat maps <v_k, v_k(x)> for inferring the location of v = (v_k, k = 1, ..., K). While individual heat maps have multiple firing locations, they add up to the Gaussian kernel with a unique location. The global Gaussian kernel is used to regulate the metrics of different constituent blocks, who vote for the inferred position by their heat maps.

To summarize, our model, while being very simple, explains the following aspects of grid cells at the computational (not necessarily neuroscience) level: (1) hexagon grid patterns. (2) metrics or grid sizes. (3) path integral. (4) path planning. (5) error correction.

We removed the original Subsection 5.2 on learning multiple blocks in the original version. We also shortened the discussion to stay within the page limit.

We will continue to revise our paper according to your advice, and we will reply to your comments soon.

Thank you for your consideration.

---

### Author Response · Authors · 2018-11-26
**List of New Results**

Dear Reviewers,

Thank you for your very helpful reviews. We have tried our best to address all the points you have raised. Please find our detailed replies under your reviews respectively.

We have uploaded the third revision. The following is a summary of the new results we have added relative to the original submitted version.


1.  Hexagon grid patterns and metrics.

Please see Sections 5.1, 5.2, and Section B of the appendix.

By introducing a generic local kernel based on the second order Taylor expansion, we are able to learn hexagon grid patterns with explicitly defined and automatically learned metrics.


2. Theoretical analysis.

Please see Section 5.1.3 and Section A of the appendix.


3. Egocentric model that couples two grid systems.

Please see Section C of the appendix.

The model couples the grid system for head direction and the original grid system for self-position. The vector of the head direction system selects or pays attention to the matrix of the self-position system.

The head direction system can also be repurposed as a clock and timestamp system.


4. Evaluations in terms of gridness measures and non-square shapes of the region.

Please see Sections B.3 and B.2.2 of the appendix.

An interesting result is that the scale measure is proportional to the automatically learned metric.


5. Ablation studies on model assumptions.

Please see Section B.4 of the appendix.

---

### Meta-Review · Area_Chair1 · 2018-12-15

**Confidence:** 5
**Recommendation:** Accept (Poster)

**Metareview:**

The authors have presented a simple yet elegant model to learn grid-like responses to encode spatial position, relying only on relative Euclidean distances to train the model, and achieving a good path integration accuracy. The model is simpler than recent related work and uses a structure of 'disentangled blocks' to achieve multi-scale grids rather than requiring dropout or injected noise. The paper is clearly written and it is intriguing to get down to the fundamentals of the grid code. On the negative side, the section on planning does not hold up as well and makes unverifiable claims, and one reviewer suggests that this section be replaced altogether by additional analysis of the grid model. Another reviewer points out that the authors have missed an opportunity to give a theoretical perspective on their model. Although there are aspects of the work which could be improved, the AC and all reviewers are in favor of acceptance of this paper.